# Distinct mechanisms of inhibition of Kv2 potassium channels by tetraethylammonium and RY785

**Shan Zhang[1], Robyn Stix[1,2], Esam A Orabi[1], Nathan Bernhardt[1], José D Faraldo-Gómez[1]***

[1]Theoretical Molecular Biophysics Laboratory, National Heart, Lung and Blood Institute, National Institutes of Health, Bethesda, United States; [2]Molecular and Cell Biology Graduate Program, Johns Hopkins University, Baltimore, United States

## eLife Assessment

This study represents an **important** advance in our understanding of how certain inhibitors affect the behavior of voltage gated potassium channels. Robust molecular dynamics simulation and analysis methods lead to a new proposed inhibition mechanism with **convincing** strength of support. This study has considerable significance for the fields of ion channel physiology and pharmacology and could aid in development of selective inhibitors for protein targets.

**\*For correspondence:**
jfg4wrk@gmail.com

**Abstract** Voltage-gated $K^+$ channels play central roles in human physiology, in health, and disease. A repertoire of inhibitors that are both potent and specific would, therefore, be of great value. RY785 has been described as promising in this regard, as it selectively inhibits channels in the Kv2 subfamily with high potency. Its mechanism of action has not yet been determined at the molecular level, but functional studies indicate it differs from those of less specific inhibitors, such as quaternary-ammonium compounds or aminopyridines. To examine this mechanism at the single-molecule level, we have carried out a series of all-atom molecular dynamics simulations based on the structure of the Kv2.1 channel in the ion-conducting state. The simulations demonstrate both RY785 and tetraethylammonium spontaneously enter the channel interior through the cytoplasmic gate, but with distinct effects. Tetraethylammonium binds to a site adjacent to the selectivity filter, on the pore axis, thus blocking the flow of $K^+$ ions. RY785, by contrast, binds to the channel walls, off-axis, and allows $K^+$ flow while the gate remains open. This observation indicates RY785 inhibits Kv2.1 by fostering the occlusion of the gate, through a network of hydrophobic interactions therein, explaining why it also modulates the voltage-sensing mechanism of the channel, 3 nanometers away.

## Introduction

Voltage-gated potassium (Kv) channels are an important class of membrane proteins involved in a variety of crucial physiological processes, both in health and disease. These channels contribute to confer electric excitability to certain types of cells, such as neurons and cardiomyocytes, by sensing and responding to changes in the transmembrane voltage. The nature of this response varies across different types of Kv channels, but in essence, it entails a structural mechanism that results in the opening or closing of a pathway across the protein through which $K^+$ ions may flow at a fast rate. This mechanism has been the focus of numerous electrophysiological studies and structure-function analyses spanning several decades, and by now, it appears well established (*Catacuzzeno et al., 2023*; *Isacoff et al., 2013*; *Vargas et al., 2012*; *Tombola et al., 2006*).

As for other proteins of interest, inhibitors of Kv channel function have been key to advance our understanding of these systems at the molecular level (*Kalia et al., 2015*). Examples include nonspecific blockers, such as quaternary-ammonium compounds (*Choi et al., 1993*) and aminopyridines (*Armstrong and Loboda, 2001*), and polypeptide toxins isolated from animal venoms, such as stromatoxin-1 (*Escoubas et al., 2002*), hanatoxin (*Swartz and MacKinnon, 1995*), and guangxitoxin (*Herrington et al., 2006*). A range of small-molecule inhibitors have also been identified (*Wulff et al., 2009*), but translational applications have been hampered by their modest potency and poor ability to discriminate among different Kv channel types. In a recent breakthrough, however, a high-throughput screening identified an inhibitor of Kv2 channels, known as RY785, which appears promising in terms of both specificity and potency (*Herrington et al., 2011*). Subsequent electrophysiological studies of the mode of action of this compound, by *Marquis and Sack, 2022*, indicated that RY785 binds to a site in the interior of rat Kv2.1 channels that, at least in part, coincides with that used by tetraethylammonium (TEA), a well-characterized cationic blocker (*Lenaeus et al., 2005*; *Faraldo-Gómez et al., 2007*; *Goodchild et al., 2012*). Access to this site requires voltage activation of the channel, i.e., opening of the ion-permeation pathway on its cytosolic side. Unlike TEA, though, RY785 is electroneutral, and so it is not immediately apparent whether it impedes $K^+$ flow directly or indirectly. Intriguingly, Sack and coworkers also report RY785 accelerates the deactivation of the voltage sensors (*Marquis and Sack, 2022*), which lie over 3 nm away from the TEA binding site. This observation indicates RY785 recognition alters the conformational energetics of the channel, and that its mode of action, as yet unknown, is more complex than that of open-pore blockers such as TEA.

In parallel, Swartz and coworkers recently succeeded in determining an atomic-resolution structure of the channel domain of rat Kv2.1, uninhibited and immersed in a lipid nanodisc, using single-particle cryo-electron microscopy (cryo-EM) (*Fernández-Mariño et al., 2023*). Like other Kv channels, the structure shows Kv2.1 is a homotetramer, and that each subunit includes six transmembrane alpha-helices (S1-S6). The ion pore is at the center of this complex, formed by the assembly of the last two transmembrane helices in each subunit (and the connector in between); the extracellular half of this pore, or selectivity filter, is much narrower than the intracellular half, and features a series of $K^+$ binding sites in a configuration that indicates the filter is in a conductive state. Four voltage-sensing domains surround the pore, each formed by the first four helices in each subunit (*Fernández-Mariño et al., 2023*); an additional alpha-helix connects the sensor to the pore in each subunit, thereby coupling the voltage response of the former to the opening/closing of the latter. Interestingly, the cryo-EM experiment resolved the channel structure in a unique conformation, namely one with the four voltage sensors in the activated state; accordingly, the intracellular side of the pore domain is wide open. Thus, despite the fact that there is no voltage difference across the lipid nanodisc, the experimental condition somehow mimics the effect of membrane depolarization (i.e. positive-inside voltages), which causes this channel to be predominantly activated/open. Interestingly, Swartz and coworkers also determined a structure of a Kv2.1 mutant in which pore and voltage domains are seemingly decoupled (*Fernández-Mariño et al., 2023*). In this structure, the sensors remain activated, and the selectivity filter is unchanged, but the intracellular side of the pore, or gate, is closed; specifically, a rearrangement of helices S5 and S6 causes a hydrophobic constriction to form at the position of residues Ile405 and Pro406, which seems too narrow to permit $K^+$ flux.

In this study, we leverage the discovery of the atomic structure of activated Kv2.1 to gain novel insights into the mode of inhibition of RY785. To do so, we employ atomically detailed computer simulations of the recognition of this inhibitor by the channel, and quantify its effect on $K^+$ permeation, relative to the uninhibited channel; the results are compared with an analogous analysis of the mode of TEA binding and inhibition, which serve as a control. Taken together, our observations indicate that RY785 inhibits Kv2.1 function not by directly blocking $K^+$ currents, when the pore is open, but by facilitating the closure of its intracellular gate.

## Results

### Simulation of activated Kv2.1 provides a realistic description of outward $K^+$ current

To establish a baseline for this investigation, we first examined the rate and mechanism of $K^+$ permeation through activated Kv2.1 in the absence of any inhibitors. To do so, we carried out an all-atom

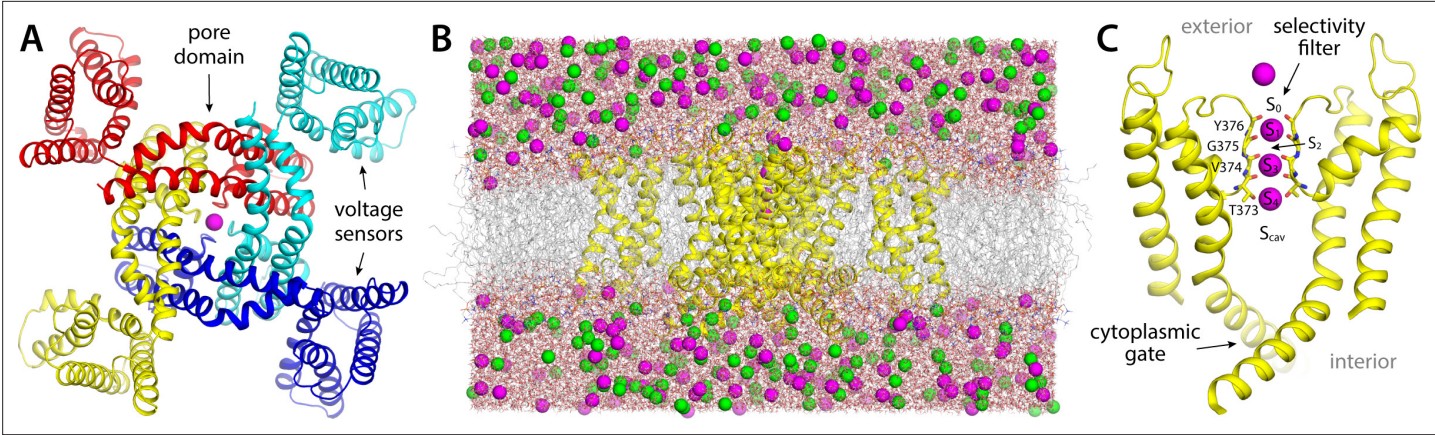

**Figure 1.** Structure and simulation of the Kv2.1 channel in the activated state. (**A**) Structure of Kv2.1 determined by single-particle cryo-electron microscopy in a lipid nanodisc (*Fernández-Mariño et al., 2023*), viewed from the cell intracellular side and along an axis perpendicular to the membrane. The channel is an assembly of four identical subunits (in colors); the transmembrane ion pore is formed at the center of the assembly, and the voltage sensors are found in the periphery, in a domain-swapped configuration. A K$^+$ ion is shown in magenta, inside the selectivity filter. (**B**) Simulation system used in this study, comprising the channel (yellow), a phospholipid bilayer (gray), and a 300 mM KCl buffer (magenta, green). The figure depicts the final configuration of the 25 µs molecular dynamics (MD) trajectory described in *Figure 2*. The total number of atoms is 201,954, some of which are omitted in the figure, for clarity. (**C**) Close-up of the pore domain and the selectivity filter. Note two protein subunits are omitted, for clarity. The configuration represented is that shown in panel (**B**); in this configuration, K$^+$ ions are found in sites $S_1$, $S_3$, and $S_4$ within the selectivity filter, while sites $S_0$, $S_2$, and $S_{cav}$ are transiently vacant.

molecular dynamics (MD) simulation of the channel embedded in a model phospholipid bilayer, under the influence of a transmembrane electric field (*Figure 1*). The channel structure employed in this calculation was that recently determined by Swartz and coworkers using cryo-EM (*Fernández-Mariño et al., 2023*); this structure comprises the pore domain and four voltage sensors (*Figure 1*), solubilized in a lipid nanodisc. To maximize the statistical significance of our results, the simulation was calculated using an Anton 2 supercomputer, which enabled us to generate a continuous 25 µs trajectory. The simulation was initialized with two K$^+$ ions inside the narrowest portion of the pore, known as the selectivity filter, and no voltage difference across the membrane; after 500 ns, a voltage difference of 100 mV was introduced, positive on the inside. As shown in *Figure 2A*, at this point K$^+$ ions begin to steadily flow across the channel in a single file and in an outward direction. In the observed mechanism, three K$^+$ ions concurrently reside in the filter, fluctuating in a concerted fashion among five transient binding sites therein, termed $S_0$, $S_1$, $S_2$, $S_3$, and $S_4$ (*Figure 2A and B*). Arrival of an additional K$^+$ ion through the cytoplasmic gate at a site termed $S_{cav}$, near the center of a water-filled cavity adjacent to $S_4$, leads to a metastable four-ion configuration. This configuration is short-lived, on account of the increased electrostatic repulsion between the K$^+$ ions, and the incoming K$^+$ ion often returns to the interior (not shown). An alternative outcome, fostered by the applied electric field, is the ejection of the outermost K$^+$ ion from the filter into the extracellular space. This event is immediately followed by the outward movement of the two other ions within the filter, and that in $S_{cav}$, thereby restoring the initial three-ion configuration (*Figure 2B*). Four iterations of this molecular cycle thus result in a complete ion permeation event. As has been noted elsewhere, water molecules can co-permeate with K$^+$ but do so rarely; in most of the observed permeation events, the ions become completely dehydrated near the center of the selectivity filter (*Köpfer et al., 2014*; *Stix et al., 2023*).

In total, we observed 45 complete permeation events, i.e., approximately one K$^+$ ion traverses the channel from side to side every 0.5 µs. This high-throughput rate appears to be explained by the fact that K$^+$ remains at near bulk density throughout the pore domain (*Figure 2C*); as K$^+$ ions cross the channel, they encounter both free-energy wells and free-energy barriers, but it appears that the balance of intermolecular interactions therein is such that these features are relatively shallow and therefore do not cause large density variations, e.g., 100-fold or greater (*Figure 2C*). Experimental estimates of the single-channel conductance of Kv2.1 range from 8 to 10 pS depending on the precise condition and type of measurement (*Pascual et al., 1997*; *Chapman et al., 2001*; *Trapani et al., 2006*); by comparison, the permeation rate observed in our simulation translates into 3 pS. Given the

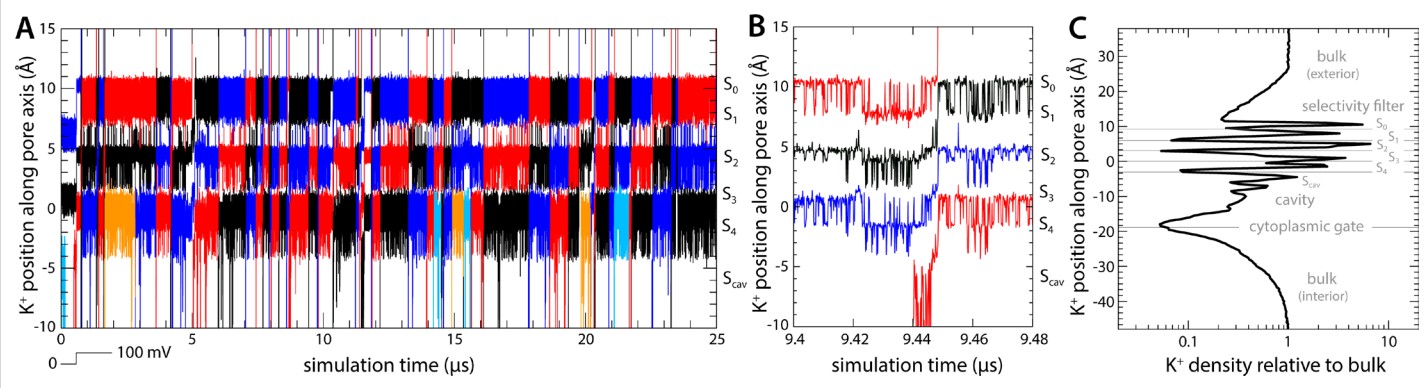

**Figure 2.** Mechanism of K$^+$ permeation through the Kv2.1 channel. (**A**) Time traces of the position of K$^+$ ions along the central axis of the channel as they reach and permeate the selectivity filter toward the extracellular side (black, red, and blue). Ions that do not reach the filter are not shown for clarity (those that reach the filter but return to the cytoplasmic side before permeating are shown in orange and cyan). The approximate location of each of the K$^+$ binding sites (S$_0$ through S$_4$, and S$_{cav}$) along the pore is indicated alongside the plot. (**B**) Same as (**A**), for a fragment of the trajectory that illustrates the knock-on mechanism that initiates and completes each of the observed permeation events. (**C**) Density for K$^+$ ions along the channel axis, relative to the bulk concentration value of 300 mM. Density peaks correspond to each of the K$^+$ binding sites within and adjacent to the selectivity filter. For reference, gray horizontal lines indicate the average position of selected protein atoms: in the selectivity filter these are, from top to bottom, the backbone carbonyl oxygens of residues Y376, G375, V374, and T373, and the side chain hydroxyl oxygen of T373; and in the cytoplasmic gate, the alpha-carbon of residue P408. All ions in the simulation system contribute to this profile, but only while they reside in a cylindrical volume of diameter equal to 12 Å, centered in and parallel to the channel axis, extending across the whole system.

myriad approximations and simplifications inherent to MD simulations, this result is very encouraging and suggests our methodology results in an adequate description of the mechanism of ion conduction of this channel. We therefore posit, by extension, that the same simulation design should be appropriate to examine how this mechanism might be modulated by an inhibitor, as described below.

## Spontaneous binding of TEA blocks K$^+$ permeation

TEA is a well-characterized nonspecific K$^+$ channel blocker; it can inhibit K$^+$ currents completely, whether applied externally or internally, albeit with modest potency. For Kv2.1 in particular, the IC values are around 5 and 0.2 mM, respectively (*Taglialatela et al., 1991*; *Kirsch et al., 1995*). Structural and computational studies carried out two decades ago using the bacterial homolog KcsA as a model system revealed that the binding site for internal TEA (and related quaternary-ammonium blockers) is adjacent to S$_{cav}$ (*Lenaeus et al., 2005*; *Faraldo-Gómez et al., 2007*); from this finding, it is logical to infer that internal TEA enters the pore through the cytoplasmic gate and that it precludes K$^+$ ions from reaching the selectivity filter.

To verify this putative inhibitory mechanism, which to our knowledge had not been directly demonstrated before, we carried out a simulation identical to that reported in the previous section, except that TEA was added in solution on the cytoplasmic side of the lipid membrane. The inhibitor was confined to explore a wide cylindrical volume co-axial with the pore but was otherwise permitted to diffuse freely. Under an applied voltage of 100 mV, however, TEA rapidly entered the channel interior through the cytoplasmic gate and did not return to the solution in a 5 µs trajectory (*Figure 3A and B*). Once in the water-filled cavity under the selectivity filter, the inhibitor continues to be highly dynamic, but tends to reside in a site about 1.5 Å away from S$_{cav}$ (*Figure 3C*), and the nitrogen atom that carries the positive charge of the ammonium ion rarely deviates more than 2 Å from the pore axis (*Figure 4A and B*). Importantly, this simulation did not reveal a single K$^+$ permeation event following TEA binding, in spite of the sustained transmembrane voltage driving K$^+$ outward (*Figure 3B*); the inhibitor prevents cytoplasmic K$^+$ ions from reaching S$_{cav}$ (*Figure 4B*), and in doing so, it precludes the onset of the knock-on mechanism of permeation through the selectivity filter described in the previous section (*Figure 2B*).

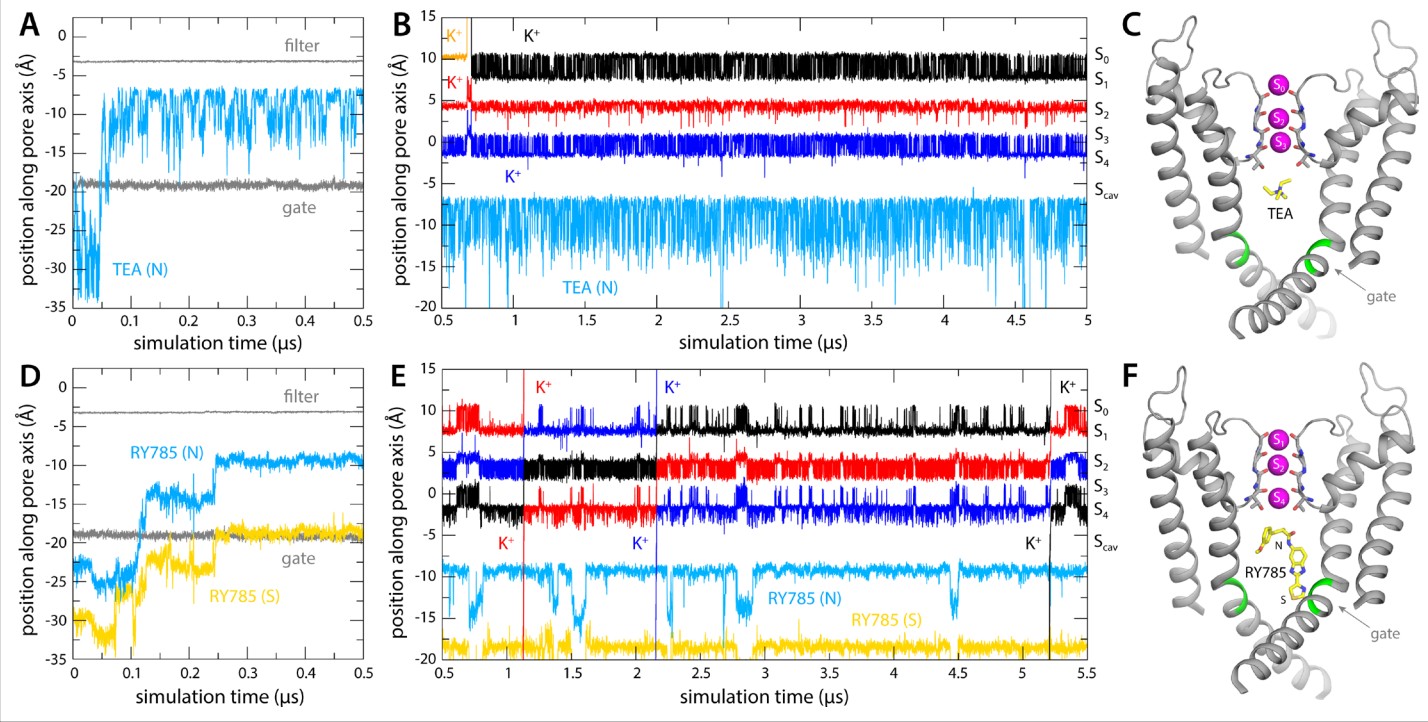

**Figure 3.** Binding of tetraethylammonium (TEA) and RY785 to Kv2.1 channel and impact on K+ permeability. (**A**) Time trace of the position of TEA (N atom) in the first 500 ns of the simulation, showing its spontaneous binding to the cavity between the selectivity filter and the cytoplasmic gate (marked by the Cα atoms of T373 and P406, respectively, indicated with gray traces). (**B**) Time trace of the position of TEA for the rest of the 5 μs simulation, alongside those for K+ ions within the selectivity filter, shown as in **Figure 2**. The location of each of the K+ binding sites therein is indicated alongside the plot. (**C**) Close-up of the channel in the final snapshot of the simulation, with TEA bound to S_cav. Only two of the four channel subunits are shown, for clarity. Residues P406 and I405 are marked in green. (**D–F**) Same as (**A–C**), for the simulation of RY785 binding, indicating separately the positions of the central N atom and of the S atom in the distal five-membered ring.

## RY785 binding to Kv2.1 permits K+ flow and involves residues in the cytoplasmic gate

As mentioned, RY785 is a specific, potent inhibitor of Kv2.1 (IC$_{50}$=50 nM) (**Herrington et al., 2011**), but the structural basis for this inhibitory activity has been unknown. Unlike TEA, RY785 is electroneutral and significantly larger, and thus it might have a very different mode of interaction with the channel. To examine this interaction and its mechanistic implications, we carried out an MD simulation identical in every way to that described above for TEA. (Note this simulation required development and optimization of a set of force field parameters for RY785, as described in Methods and in Appendix 1.) This simulation also showed that RY785 can readily penetrate the cytoplasmic gate, though more slowly than TEA (**Figure 3D**). While RY785 is not cationic, it features an electric dipole, and the 100 mV voltage difference applied across the simulation system (positive inside) orients the inhibitor as it enters the channel. Specifically, the *m*-methoxybenzene moiety passes through the gate first and is ultimately closer to the selectivity filter, while the benzimidazole and thiazole moieties, which are more electronegative, cross the gate last and thus end up being closer to the cytoplasmic space (**Figure 3D and F**). Like TEA, RY785 continued to be very dynamic inside the pore, though it primarily sampled a unique binding mode, in four different configurations reflecting the symmetry of the channel; displacements parallel to the pore axis were far less common than for TEA, and RY785 did not escape from the protein interior in a 5.5 μs trajectory (**Figure 3E and F**). Nevertheless, we observed that RY785 did not prevent the flow of K+ ions through the channel, nor did it significantly alter the knock-on mechanism within the filter (**Figure 3E**). Unlike TEA, RY785 does not prevent cytoplasmic K+ ions from reaching the S_cav site, though they clearly do so more rarely; thus, the permeation rate we observe is approximately four times slower than for the uninhibited channel (**Figure 3E**).

Further inspection of the simulation data reveals why RY785 does not block the outward flow of K+ ions. As shown in **Figure 4**, this inhibitor binds to the wall of the cavity through the benzimidazole

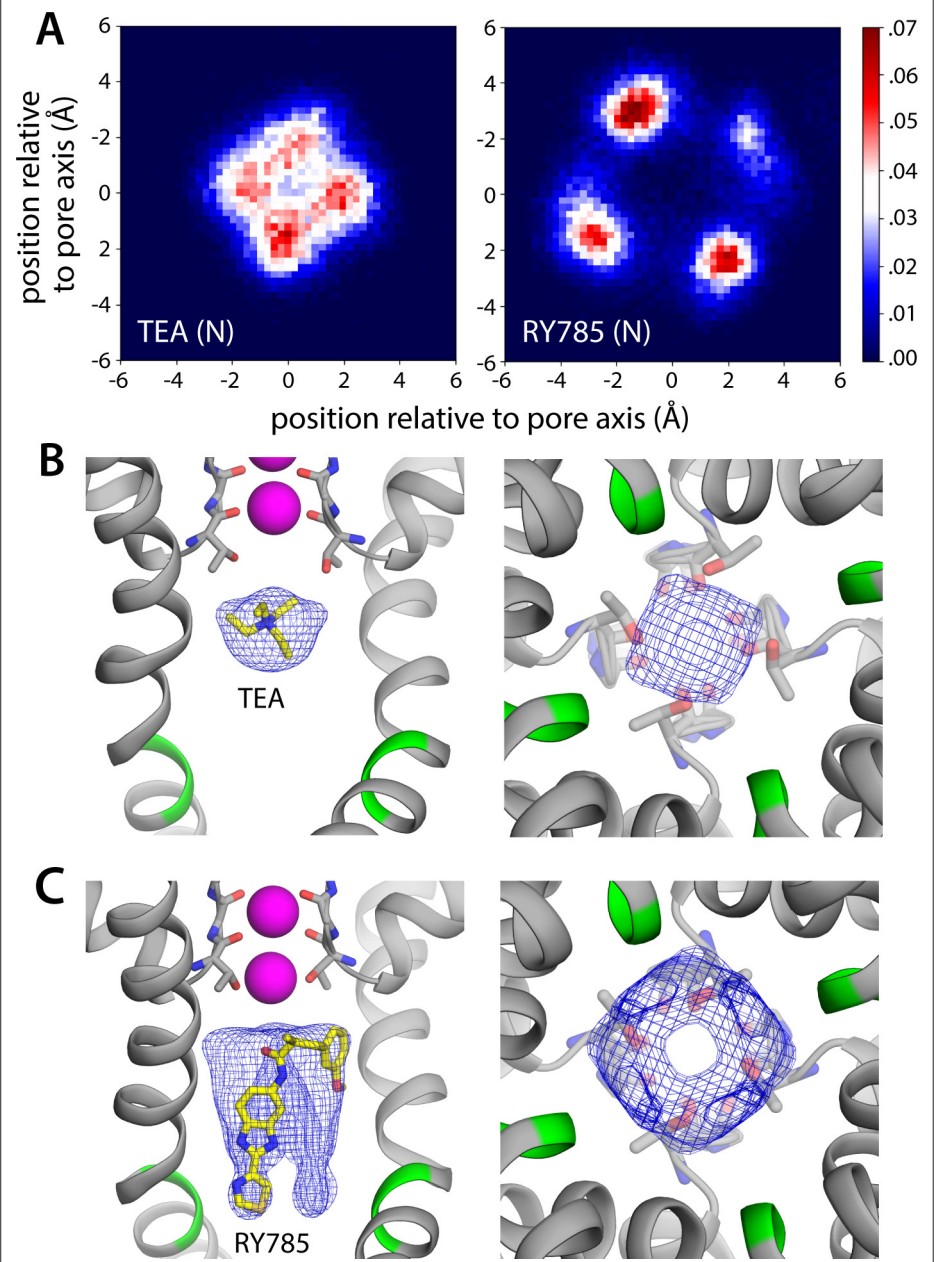

**Figure 4.** Bound RY785 does not occlude the K+ pathway, but tetraethylammonium (TEA) does. (**A**) Probability histograms for the position of TEA and RY785 (in Å) as projected on the plane of the membrane, i.e., perpendicular to the pore, with the origin on its central axis. (**B**) Snapshot of the simulation of Kv2.1 and TEA described in *Figure 3*, with a density map for TEA (blue mesh) calculated for all non-hydrogen atoms and all simulation snapshots. The map is viewed from the membrane plane, on the left, as well as from the cytoplasmic entrance of the pore, on the right. (**C**) Same as (**B**), for the simulation of Kv2.1 and RY785. All maps are contoured at the same sigma value (equal to 0.04).

and thiazole moieties; the *m*-methoxybenzene ring projects toward the center of the pore, but the high flexibility of this segment leaves an open pathway for K+ throughout the length of the cavity. Logically, this pathway is narrower than for the apo protein, explaining the slower permeation rate, but it is sufficient to permit K+ to access the $S_{cav}$ site and ultimately enter the selectivity filter. This particular mode of interaction stems from persistent hydrophobic interactions with residues in the S6 helices (*Figure 5*). In particular, the most important contacts are Val409, Pro406, Ile405, Ile401, and Val398 in one or more subunits of the tetramer; among these, Pro406, Ile401, and Val398 appear to be

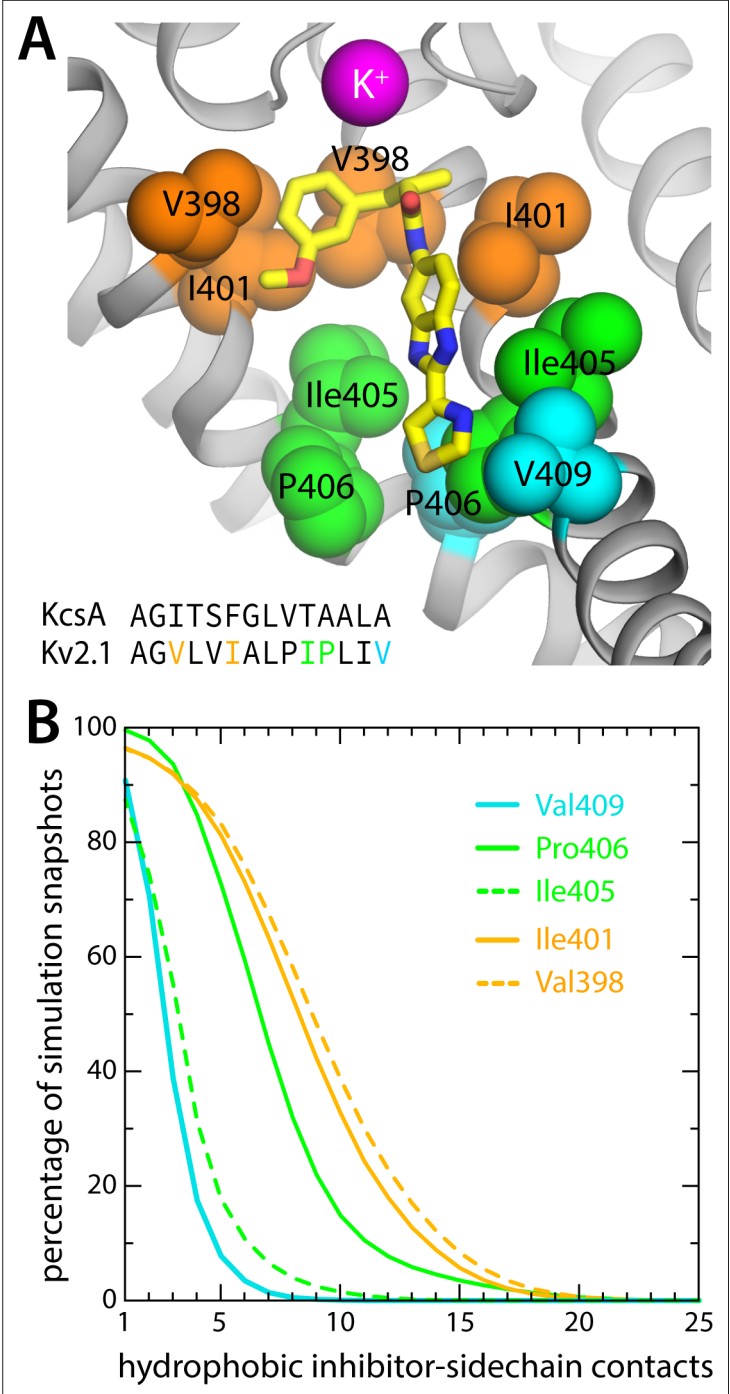

**Figure 5.** RY785 binds to the interior wall in the Kv2.1 cavity. (**A**) Close-up of the snapshot shown in *Figure 4C*, highlighting the most frequently observed protein contacts for RY785; these are on helix S6, which lines the cavity. Only two adjacent protein subunits are shown for clarity. Note that V409, P406, and Ile405 are located at the narrowest point of the permeation pathway (aside from the selectivity filter). For reference, the inset shows an alignment of the sequence of this region of S6 (residues 396–409) with its equivalent in the KcsA K$^+$ channel (residues 98–111), whose closed-state structure is known (*Zhou et al., 2001*). A more comprehensive comparison of the sequences of the S6 segment in K$^+$ and Ca$^{2+}$ channels is included in *Figure 5—figure supplement 1*. (**B**) Statistical analysis of the interactions depicted in panel (**A**). A contact was defined as an instance wherein a C atom in the abovementioned hydrophobic side chains was within 4.5 Å of one of the C/S atoms in RY785; a given snapshot might thus show multiple contacts with each side chain, sometimes in two different subunits. The plot in the figure quantifies the percentage of simulation snapshots in which a given number of contacts were observed

*Figure 5 continued on next page*

*Figure 5 continued*

for each side chain at minimum; e.g., in 80% of the snapshots RY785 forms ~5 or more contacts with Pro406; in 1/4 of those, or 20% of the total, the number of observed contacts is ~10 or more. Contacts with Val409 and Ile405 are mostly in one subunit, while Pro406, Ile401, and Val398 are contacted simultaneously in two subunits.

The online version of this article includes the following figure supplement(s) for figure 5:

**Figure supplement 1.** Multiple-sequence alignment of the S6 segment in selected K[+] and Ca[2+] channels.

key, as the inhibitor often contacts these residues simultaneously in two protein subunits (*Figure 5A*). As will be discussed below, this observation is important because several of these residues form the hydrophobic seal that closes the cytoplasmic entrance to the pore in the deactivated, closed state of the channel.

## Bound TEA and RY785 differentially impact K[+] access to the selectivity filter

To corroborate the results described above, we used an alternative simulation design wherein no voltage difference was applied across the membrane to drive K[+] outward. Instead, we induced a knock-on event within the selectivity by forcibly driving the central K[+] ion (in either $S_2$ or $S_3$) to the $S_1$ site, reasoning this perturbation would result in the immediate ejection of the outermost K[+], the movement of the innermost K[+] into $S_2$ or $S_3$, and the creation of an artificial vacancy in the $S_4$ site. With either TEA or RY785 occupying the water-filled cavity, just as in the simulations described above, we asked which inhibitor would permit or preclude reloading of site $S_4$ by a K[+] ion entering the pore through the cytoplasmic gate.

As shown in *Figure 6*, the ion dynamics within the selectivity filter were as anticipated, reproducibly and irrespective of which inhibitor is bound to the channel. The simulations with TEA and RY785 clearly differ otherwise. While TEA is bound, the $S_4$ site remains vacant after the induced knock-on event; only when TEA spontaneously diffuses back into the cytoplasm prior to the induced knock-on event (note that no electric field is applied) do we observe a K[+] ion quickly binding to $S_4$, thereby restoring

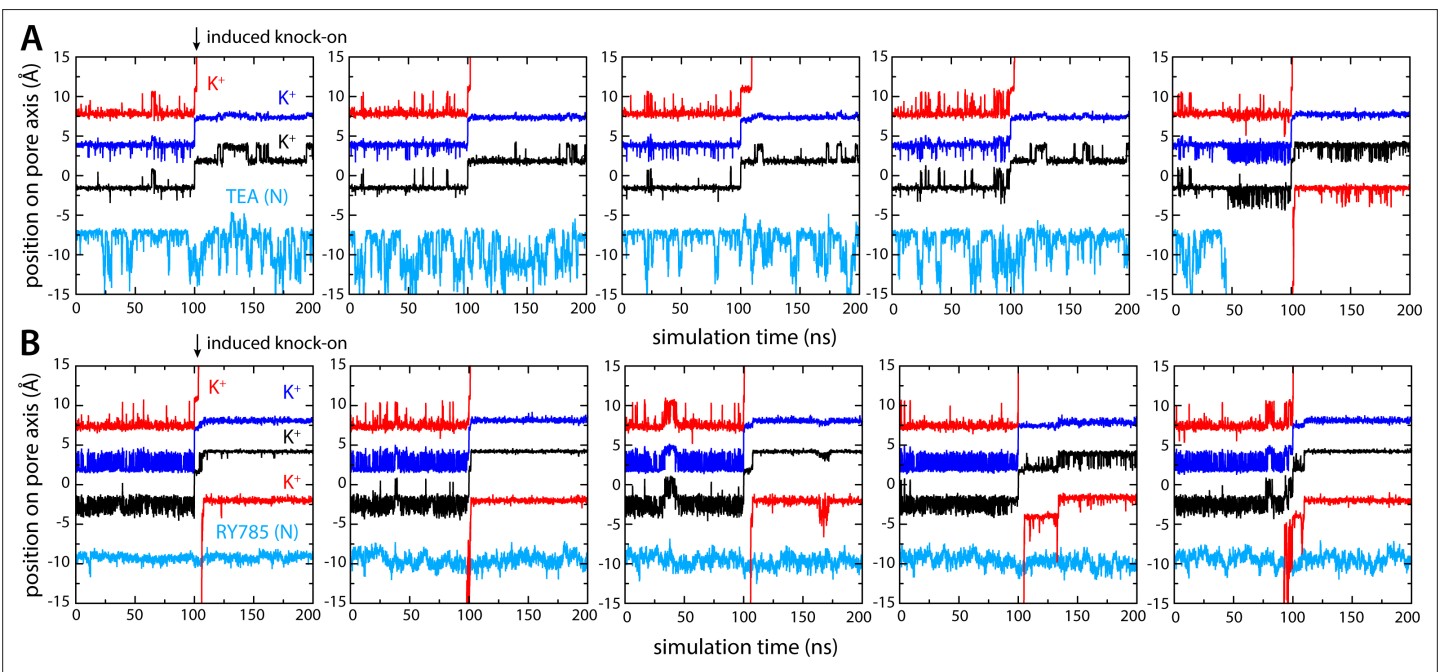

**Figure 6.** Bound tetraethylammonium (TEA) precludes K[+] from accessing the $S_4$ binding site, but bound RY785 allows it. (**A**) Five independent simulations of Kv2.1 bound to TEA wherein a knock-on event was artificially induced in the selectivity filter at $t$ = 100 ns, to create a K[+] vacancy in site $S_4$. K[+] did not reload this site in the subsequent 100 ns, except when TEA spontaneously dissociated prior to the knock-on event (right). (**B**) Same as (**A**), for Kv2.1 bound to RY785. In all simulations, a K[+] ion reloads the $S_4$ site within ~10 ns of the induced knock-on event, corroborating RY785 is not an open-state blocker of Kv2.1.

the preferred three-ion configuration (*Figure 6A*). Reloading of a K$^+$ is also what we observed in all the simulations we carried out with bound RY785, consistent with the observation this inhibitor does not physically occlude the pathway for K$^+$ into the selectivity filter in the open state of the channel (*Figure 6B*). These results are completely coherent with those obtained when voltage was used as a driving force for K$^+$ flow in demonstrating that TEA is an open-channel blocker, but RY785 is not.

## Discussion

Previous experimental studies have shown that RY785 is a potent, selective inhibitor of Kv2 channels (*Herrington et al., 2011*). Nonetheless, its site of recognition and precise mechanism of action have been unknown. Allosteric inhibitors featuring thiazole moieties, like RY785 does, have been shown to interact with the voltage-sensing domains of several voltage-gated Na$^+$ channels (*McCormack et al., 2013*; *Ahuja et al., 2015*), suggesting RY785 could act similarly. However, available electrophysiological evidence shows that RY785 competes with TEA, a well-studied K$^+$ channel inhibitor known to bind within the pore domain (*Marquis and Sack, 2022*); it seems very probable therefore that the binding sites for these two inhibitors coincide, at least in part. The computer simulations of Kv2.1 presented in this study were designed to support or refute the hypothesis that RY785 binds to the pore domain, without other a priori assumptions. Ultimately, the results support that notion: when the cytoplasmic gate of the channel is open, both RY785 and TEA can enter the K$^+$ permeation pathway (though not simultaneously) and ultimately dwell in a binding site within the water-filled cavity that precedes the selectivity filter. Aside from this similarity, however, the mechanisms of action of these two inhibitors seem entirely different.

TEA is a small tetrahedral molecule and is positively charged, though it is larger than a K$^+$ ion. Thus, under positive-inside voltages, it quickly enters the ion permeation pathway and binds near the entrance of the selectivity filter, which is too narrow for TEA to penetrate. Because the molecular symmetry of TEA does not match the fourfold symmetry of the channel, TEA tends to dwell slightly off-axis; however, this deflection is smaller than the size of a K$^+$ ion, so on average the blocker appears to be on the channel axis. Consistent with its location and charge, we observe that TEA completely blocks the outward permeation of K$^+$ ions under our simulation conditions; in contrast, in an independent simulation under the same conditions, but in the absence of TEA, we observe instead fast K$^+$ throughput, at a rate comparable to experimental measurements. Comparison of these two simulations shows that TEA blocks K$^+$ currents because it prevents K$^+$ ions in the cytoplasmic space from destabilizing the K$^+$ ions that reside in the selectivity filter.

RY785, however, does not block the flow of K$^+$ ions through the channel while the cytoplasmic gate is open. RY785 is electroneutral and mostly hydrophobic; thus, despite being larger than TEA, RY785 does not tend to occupy the center of the pore. Instead, it packs against the walls of the cavity, forming interactions with a cluster of hydrophobic side chains that, collectively, is specific to Kv2 channels (*Figure 5*, *Figure 5—figure supplement 1*). Recognition of RY785 therefore does not occlude the pathway for K$^+$ into the selectivity filter. How does RY785 inhibit Kv2 channels then? A puzzling but revealing experimental observation is that RY785 recognition by Kv2.1 channels, following activation, accelerates their deactivation (*Marquis and Sack, 2022*). That is, RY785 appears to trap itself within the cavity upon closure of the extracellular gate. Under normal conditions, gate closure requires the repolarization of the membrane, i.e., an energy input. Negative-inside voltages drive the S4 helix in each of the voltage sensors to the downstate (creating the so-called gating currents), and this motion is communicated to the pore domain via the S4-S5 linker, ultimately causing a rearrangement in helices S5 and S6, which closes the cytoplasmic gate (*Figure 7*). Clearly, from its binding site in the interior of the pore domain, RY785 cannot directly impact the voltage sensors, nor their mechanical coupling with S5-S6. Nonetheless, our simulations indicate an indirect effect. We observe that RY785 forms persistent interactions with the hydrophobic residues that are likely to constrict the pore in the closed state, in the S6 helix (*Fernández-Mariño et al., 2023*); in several cases, RY785 interacts simultaneously with the same residue in two adjacent channel subunits. These observations suggest that RY785 might stabilize a semi-closed conformation of the gate that is no longer permeable to K$^+$, even though the structural rearrangement in S5-S6 that is required for full closing has not been achieved completely (*Figure 7*). That is, an opening between the four protein subunits would remain at the position of the gate, but this opening would be occupied by RY785, bridging hydrophobic interactions between protein residues therein. Importantly, this hypothetical conformation might not

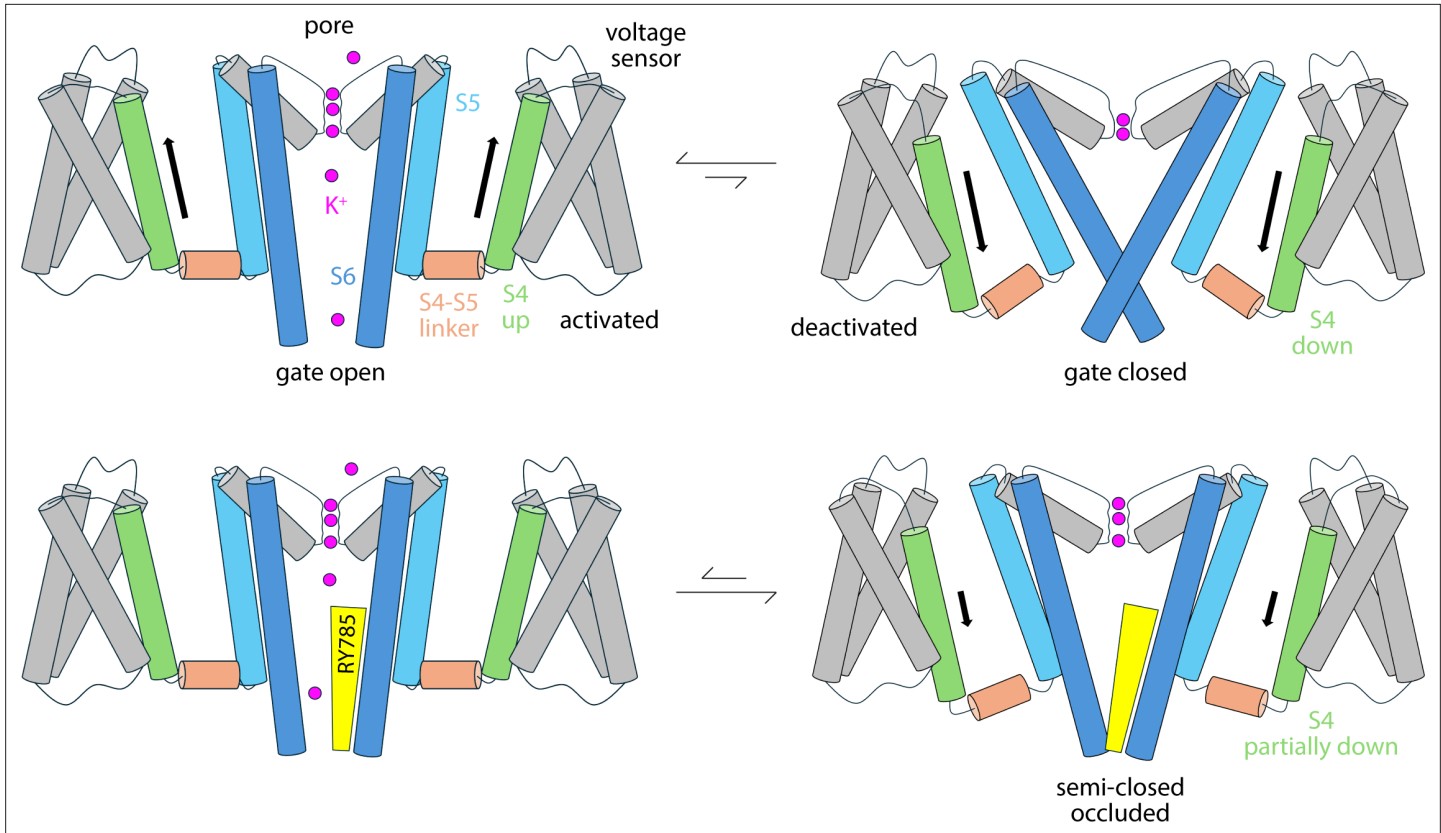

**Figure 7.** Hypothetical mode of Kv2 channel inhibition by RY785. R785 does not directly block K$^+$ flow; instead, it reshapes the conformational free-energy landscape of the pore domain to stabilize an occluded, partially closed state that does not require S4 to reach the downstate in full. The model is an inference based on the simulation data reported in this article and a previous electrophysiological study by **Marquis and Sack, 2022**. Note that this is a simplified diagram of the functional cycle of the channel, which includes nonconductive or transiently inactivated states, aside from the fully deactivated form (**Stix et al., 2023**; **Tan et al., 2022**).

require the voltage sensors to reach the downstate completely (**Figure 7**); if so, K$^+$ currents would be terminated more readily upon repolarization with bound RY785 than otherwise. This model would also explain the experimental observation that RY785 partially inhibits the current generated by the motion of the S4 helix, known as gating current (**Marquis and Sack, 2022**), even though S4 is about 3 nm away from the RY785 binding site. That is, while this study is not specifically designed to evaluate the possibility that RY785 binds directly to the voltage sensors, the mechanism we propose would explain the observed effects of RY785 recognition on the gating currents.

While further studies will be required to fully validate or refute this mechanistic model, recently published work supports our hypothesis. After this study was first reported in 2024, the structure of a Kv2.1 channel with a closed pore domain was successfully determined (**Mandala and MacKinnon, 2025**). That structure shows that a Pro-Ile-Pro motif in helix S6 marks the position of the intracellular gate, where the pore domain constricts maximally (aside from the selectivity filter). As illustrated in **Figure 5**, this motif is precisely where the benzimidazole and thiazole moieties of RY785 are recognized in our simulations. The inhibition mechanism we outline in **Figure 7** thus seems increasingly plausible. Nonetheless, the fact that both open and closed states of the pore are now known provides an excellent opportunity to examine this model further. Specifically, it is now possible to map the intrinsic conformational free-energy landscape of the pore domain, in a simplified construct lacking the voltage sensors, using enhanced-sampling simulations (**Marinelli and Faraldo-Gomez, 2024**); we anticipate that both the open and closed states of the gate will be identifiable free-energy minima in this landscape, and that the closed state will be less favorable than the open state (in the absence of the voltage sensors). Comparison of analogous calculations carried out with and without bound RY785, alongside an evaluation of the changes in K$^+$ permeability in each case (**Oh et al., 2022**), would

provide further quantitative insights into the intriguing mechanism by which RY785 inhibits gating and ion flow in Kv2 channels and, in turn, aid future optimizations of this inhibitor.

## Methods

### MD simulations of K$^+$ flux through Kv2.1 in 100 mV

The MD simulations of activated wild-type Kv2.1 are based on the cryo-EM structure of the activated, open state (PDB entry 8SD3) (*Tan et al., 2022*). In all cases, the simulations examined a construct encompassing residues 174–426 (S1-S6), with neutral N- and C-termini and all ionizable side chains in their default protonation state at pH 7, which in this construct results in a net charge of zero. Two K$^+$ ions were initially positioned in the selectivity filter, one coordinated by residues 373 and 374 (site S$_3$) and another by residues 375 and 376 (site S$_1$) (*Figure 1C*); a third ion was positioned below the side chain of 373, outside the filter, and two water molecules were modeled between the three ions. (This configuration was very short-lived and replaced by others in which three K$^+$ ions concurrently occupy the filter, typically without water molecules in between – see Results.) For all systems, we used Dowser (*Zhang and Hermans, 1996*) to model structural water molecules within the protein that were not resolved in the experimental density maps. To complete the initial simulation setup of the experimental cryo-EM structure, the construct including ions and water molecules was energy-minimized using CHARMM (*Brooks et al., 2009*) and the CHARMM36m force field (*Huang et al., 2017*; *Best et al., 2012*; *Klauda et al., 2010*); specifically, the minimization used the steepest-descent algorithm for 250 steps, and then conjugate-gradient algorithm for another 250 steps.

The channel was simulated in a POPC lipid bilayer and a 300 mM KCl solution. (Note that while this concentration is atypical in the intracellular environment, it is often used in electrophysiological studies. In the simulation context, this choice is sensible in that it leads to a greater number of permeation events, and therefore, greater statistical confidence in the calculated ion conductance, or its inhibition.) To generate a molecular model of this membrane/solvent environment, we created a coarse-grained (CG) POPC-lipid bilayer in 300 mM KCl in an orthorhombic box of ~150×150×100 Å using insane.py (*Wassenaar et al., 2015*). To equilibrate this CG system, we carried out a 20 μs MD simulation using GROMACS 2018.8 (*Pronk et al., 2013*) and the MARTINI 2.2 force field (*de Jong et al., 2013*; *Marrink et al., 2007*) at constant temperature (303 K) and constant semi-isotropic pressure (1 atm) and with periodic boundary conditions. The integration time-step was 20 fs. We then embedded the structure of the Kv2.1 channel in this environment; to do so, the atomic structure of the channel was first coarse-grained with martinize.py (*de Jong et al., 2013*) and overlaid onto the equilibrated membrane/solvent system, removing all overlapping lipid and water molecules. Then, to optimize the resulting protein/lipid/solvent interfaces, we carried out a 10 μs MD simulation of the complete system using GROMACS 2018.8 (*Pronk et al., 2013*) and MARTINI 2.2 (*de Jong et al., 2013*; *Marrink et al., 2007*) at constant temperature (303 K) and constant semi-isotropic pressure (1 atm), with periodic boundary conditions and an integration time-step of 20 fs. Having ascertained the equilibration of the membrane structure and of the position and orientation of the protein in the lipid bilayer, the final snapshot of the CG trajectory was transformed into an all-atom representation compatible with the all-atom CHARMM36m force field (*Huang et al., 2017*; *Best et al., 2012*; *Klauda et al., 2010*). To do so, lipid and solvent molecules were back-mapped onto all-atom models (*Wassenaar et al., 2014*), while the CG version of Kv2.1 was replaced (not back-mapped) with the energy-minimized all-atom construct described above, after optimally superposing the Cα trace of the latter onto that of the former. The resulting all-atom molecular system includes 526 POPC lipids, 38,629 water molecules, and 208 K$^+$ and 239 Cl$^-$ (300 mM KCl) in an orthorhombic box of ca. 150×150×100 Å. To further optimize this all-atom model, the simulation system was first energy-minimized for 5000 steps with NAMD 2.13 (*Phillips et al., 2020*; *Fiorin et al., 2013*) and CHARMM36m (*Huang et al., 2017*; *Best et al., 2012*; *Klauda et al., 2010*), using the conjugate-gradient algorithm. We then carried out a series of MD simulations wherein structural restraints applied to the protein and ions/water within the selectivity filter are progressively weakened over the course of ~150 ns, as previously described (*Stix et al., 2023*; *Tan et al., 2022*). These simulations were carried out using NAMD 2.13 (*Phillips et al., 2020*; *Fiorin et al., 2013*) and CHARMM36m (*Huang et al., 2017*; *Best et al., 2012*; *Klauda et al., 2010*) at constant temperature (298 K) and constant semi-isotropic pressure (1 atm) with periodic boundary conditions and an integration time-step of 2 fs. Electrostatic interactions were calculated using the

Particle-Mesh Ewald method (**Darden et al., 1993**), with a real-space cutoff value of 12 Å; van der Waals interactions were also cut off at 12 Å, with a smooth switching function taking effect at 10 Å.

To quantify the ion-conducting properties of activated Kv2.1, a 25 μs MD trajectory was calculated under an applied voltage (positive inside) using an Anton 2 supercomputer (**Shaw et al., 2014**) and the all-atom CHARMM36m force field (**Huang et al., 2017**; **Best et al., 2012**; **Klauda et al., 2010**). The starting configuration for this simulation was the final configuration of the abovementioned equilibration process. The simulation was again carried out at constant temperature (298 K) and semi-isotropic pressure (1 atm), using the Nosé-Hoover thermostat (**Nosé, 1984**; **Hoover, 1985**) and the Martyna-Tobias-Klein barostat (**Martyna et al., 1994**), respectively, with periodic boundary conditions and an integration time-step of 2.5 fs. Electrostatic interactions were calculated using the Gaussian-split Ewald method (**Shan et al., 2005**); van der Waals interactions were cut off at 10 Å. To preclude large-scale changes in the secondary or tertiary structure of the channel that might develop in the 25 μs timescale, due to cumulative force field inaccuracies, the energy function of the simulation was supplemented with a weak biasing potential that favors the known experimental geometry, while permitting the structure to locally fluctuate as required for ligand recognition and ion permeation (**Stix et al., 2023**; **Tan et al., 2022**). This potential acts on all $\phi$, $\psi$, and $\chi_1$ dihedral angles in the protein and is defined by:

$$U\left(\theta_t\right) = k \sum_{m=1}^{m=6} (-1)^m \left[1 + \cos(m\theta_t - m(\theta_{expt} - 180))\right]/m!$$

where $\theta_t$ is the value of each dihedral angle at time $t$ in the simulation, $\theta_{expt}$ denotes the corresponding value in the experimental structure, and $k = 1k_BT$. Note this potential is identical to that used in previous studies based on multi-microsecond Anton 2 simulations (**Pan et al., 2019**; **Jensen et al., 2012**; **Park et al., 2023**), except the magnitude of the bias in this study is considerably weaker (i.e. smaller $k$). To drive $K^+$ ions outward, a transmembrane voltage difference was applied across the membrane, positive inside. This voltage difference was applied as a jump from 0 to 100 mV at timepoint 0.5 μs and sustained thereafter. The desired voltage resulted from the application of an outwardly directed, constant electric field perpendicularly to the membrane plane (**Gumbart et al., 2012**). For a simulation box of approximately 92 Å in that direction, a voltage difference of 100 mV corresponds to an electric field of 0.024 kcal mol$^{-1}$ Å$^{-1}$ e$^{-1}$ (note that 1 kcal mol$^{-1}$ Å$^{-1}$ e$^{-1}$ = 43.4 mV/Å).

## Simulations of TEA and RY785 binding and inhibition in 100 mV

Additional MD simulations were carried out to examine the mode of TEA and RY785 binding and inhibition. These simulations used, as a starting point, the final configuration of the 25 μs MD trajectory described above; a molecule of TEA or RY785 was added about 10 Å below the cytoplasmic gate, and overlapping water molecules were removed. These constructs were then energy-minimized using CHARMM (**Brooks et al., 2009**) and the CHARMM36m force field (**Huang et al., 2017**; **Best et al., 2012**; **Klauda et al., 2010**); specifically, the minimization consisted of 100 steps using the steepest-descent algorithm with the protein and $K^+$ ions within fixed. To further optimize these all-atom models, the simulation systems were then energy-minimized for 5000 steps with NAMD 2.13 (**Phillips et al., 2020**; **Fiorin et al., 2013**) and the CHARMM36m force field (**Huang et al., 2017**; **Best et al., 2012**; **Klauda et al., 2010**), using the conjugate-gradient algorithm. Again using NAMD 2.13, we then carried out a 20 ns MD simulation of each channel-inhibitor system wherein weak structural restraints identical to those used in the 25 μs MD trajectory were applied to the protein. In addition, a flat-bottom confining potential was applied to TEA or RY785 to keep the inhibitor inside a cylindrical volume co-axial with the channel pore, 20 Å in diameter. Specifically, we used atoms Y376:C$_\beta$ and V409:C$_\beta$ in the four protein subunits to define this axis, and the inhibitor was confined by a potential of the form:

$$U\left(d_t\right) = 0.5k\left(d_t - d_o\right)^2 \; if \, d_t > d_o$$
$$U\left(d_t\right) = 0 \; if \, d_t \leq d_o$$

where $d_t$ is the distance between the center-of-mass of the inhibitor and the channel axis, perpendicularly to that axis, $d_o$ is 10 Å and $k = 100$ kcal/mol/Å$^2$. To preclude the inhibitor from diffusing across the periodic boundary of the system (i.e. preserving it in the cytoplasmic side of the channel),

a confining potential of the same functional form was also applied relative to the center of the four V409:$C_\beta$ atoms; in this case, $d_t$ is the projection of the distance onto the channel axis, while $d_o$ is 15 Å. Following these 20 ns equilibrations, trajectories of 5 and 5.5 µs were calculated on Anton 2 for the TEA and RY785 systems, respectively, using exactly the same settings as those reported above for the channel-only simulation.

## Simulations of TEA and RY785 inhibition after induced K⁺ knock-on event

The simulations described in *Figure 6* used, as starting point, the final configurations of the trajectories calculated with Anton 2 for the channel-inhibitor systems (*Figure 3*). In these starting configurations, three K⁺ ions occupy the selectivity filter, and TEA or RY785 resides within the cavity. To create a vacancy in the $S_4$ site, without a voltage difference across the membrane, the central K⁺ ion (which fluctuates between $S_2$ and $S_3$) was driven outward at $t=100$ ns, by activating a set of restraints that bring the ion within coordination distance of the carbonyl O atoms of Y376 and G375 in all four subunits (ca. 3 Å, $k=3$ kcal/mol/Å²). As a result, the outermost K⁺ ion (which fluctuates between $S_0$ and $S_1$) was ejected, and the innermost K⁺ ion moved to the center of the filter, creating the desired vacancy in $S_4$. The simulations were then continued for 100 ns to examine whether or not K⁺ ions would reload this vacancy. These simulations were carried out using NAMD 3.0b6 (*Phillips et al., 2020*; *Fiorin et al., 2013*) and the CHARMM36m force field (*Huang et al., 2017*; *Best et al., 2012*; *Klauda et al., 2010*). All other settings, including confining potentials, were as described above.

## Development of molecular-mechanics force field for RY785

To our knowledge, no molecular-mechanics (MM) force field for RY785 was publicly available prior to this work. Thus, a new force field had to be developed and optimized to be able to introduce this molecule in our simulations of Kv2.1 – specifically one compatible with the CHARMM General Force Field, or CGenFF (*Vanommeslaeghe and MacKerell, 2012*). Following recommended procedures (*Vanommeslaeghe and MacKerell, 2012*; *Vanommeslaeghe et al., 2010*; *Vanommeslaeghe et al., 2012*), we carried out a series of quantum chemical calculations to generate ab initio benchmarking data, including atomic charges, optimized molecular geometries, and multiple potential energy surfaces (PESs) generated by relaxed scan of selected bond and dihedral angles; these calculations were performed in the gas phase using Gaussian 16 (Gaussian, Inc, Wallingford, CT, 2016), using the MP2/6-31G(d) method. Structural properties and calculated PESs were then used to adjust atomic-charge, bond-angle, and dihedral-angle parameters lacking adequate entries in CGenFF. These MM parameters were refined targeting the benchmark QM data, including interaction energies and H-bond distances for multiple RY785-$H_2O$ binary complexes. Each of these RY785-$H_2O$ conformers probes the interaction of a $H_2O$ molecule with a different H-bond donor (C-H or N-H) or acceptor (N or S) in RY785. The complexes were set up with an ideal H-bond interaction between RY785, in its MP2/6-31G(d) optimized geometry, and a water molecule, in its idealized TIP3P geometry (*Jorgensen et al., 1983*). The interaction distances in these complexes were optimized at the HF/6-31G(d) level while keeping all other degrees of freedom fixed; the interaction energy was then calculated without correction for basis set superposition error and multiplied by a factor of 1.16, as suggested elsewhere (*Vanommeslaeghe et al., 2012*). Further details can be found in Appendix 1.

## Acknowledgements

This research was funded by the intramural program of the National Heart, Lung and Blood Institute, National Institutes of Health (NIH). Computational resources were provided by the NIH High-Performance Computing Facility 'Biowulf' and by the Pittsburgh Supercomputing Center, which provided access to an Anton 2 computer donated by D.E. Shaw Research and supported through NIH grant R01GM116961.

## Additional information

### Funding

| Funder | Grant reference number | Author |
|--------|------------------------|--------|
| NIH | NHLBI Intramural | Shan Zhang<br>Robyn Stix<br>Esam A Orabi<br>Nathan Bernhardt<br>José D Faraldo-Gómez |

The funders had no role in study design, data collection and interpretation, or the decision to submit the work for publication.

### Author contributions

Shan Zhang, Formal analysis, Investigation, Visualization, Writing – original draft, Writing – review and editing; Robyn Stix, Formal analysis, Investigation, Methodology, Writing – original draft; Esam A Orabi, Formal analysis, Investigation, Visualization, Methodology, Writing – original draft, Writing – review and editing; Nathan Bernhardt, Supervision, Methodology; José D Faraldo-Gómez, Conceptualization, Resources, Formal analysis, Supervision, Funding acquisition, Visualization, Writing – original draft, Project administration, Writing – review and editing

### Author ORCIDs

Esam A Orabi ⬤ https://orcid.org/0000-0002-5338-9203
José D Faraldo-Gómez ⬤ https://orcid.org/0000-0001-7224-7676

Reviewer #3 (Public review): https://doi.org/10.7554/eLife.101855.4.sa1
Author response https://doi.org/10.7554/eLife.101855.4.sa2

## Additional files

### Supplementary files

MDAR checklist

### Data availability

Forcefield parameter files for RY785 can be downloaded from https://github.com/Faraldo-Gomez-Lab-at-NIH/Download (copy archived at *Faraldo-Gómez, 2026*).

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

## Appendix 1

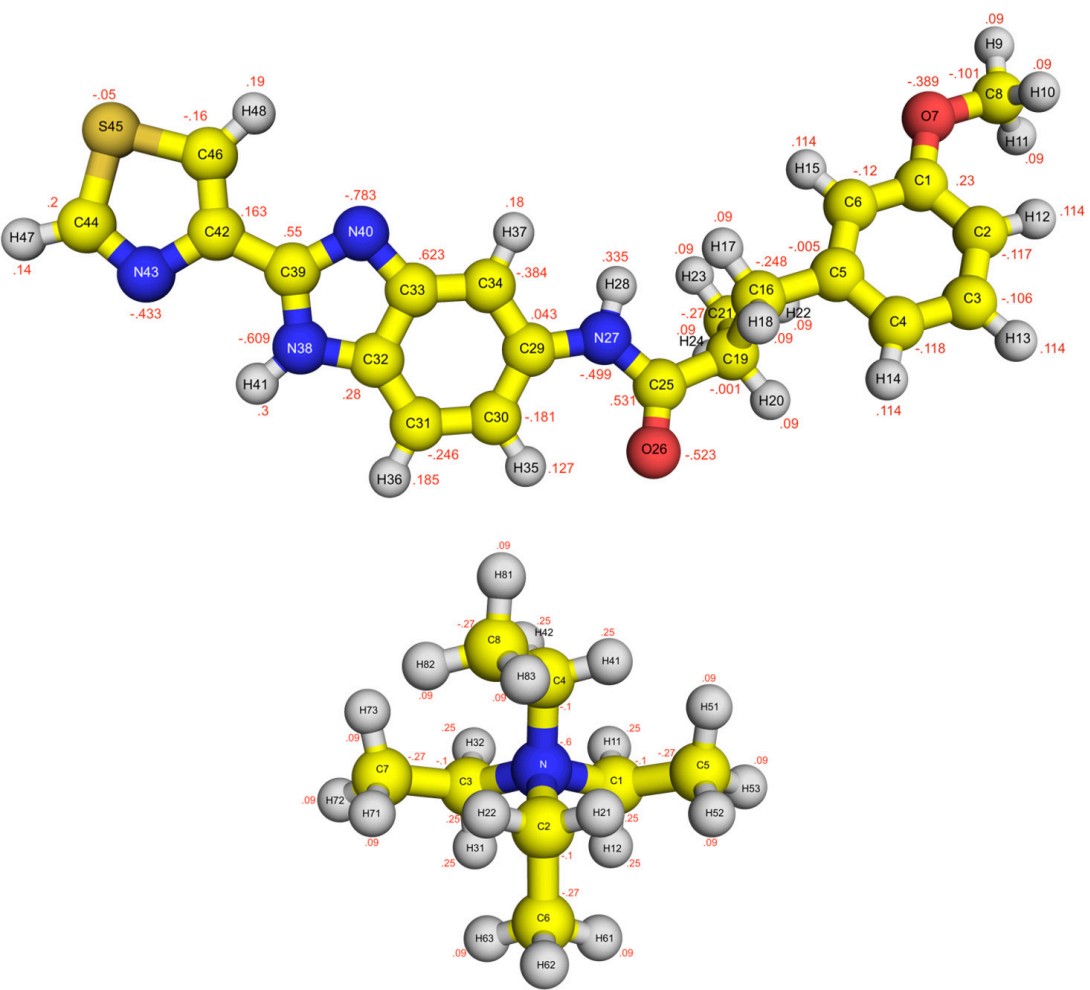

**Appendix 1—figure 1.** Chemical structures of (top) RY785 and (bottom) tetraethylammonium. Atom names and partial electronic charges are indicated.

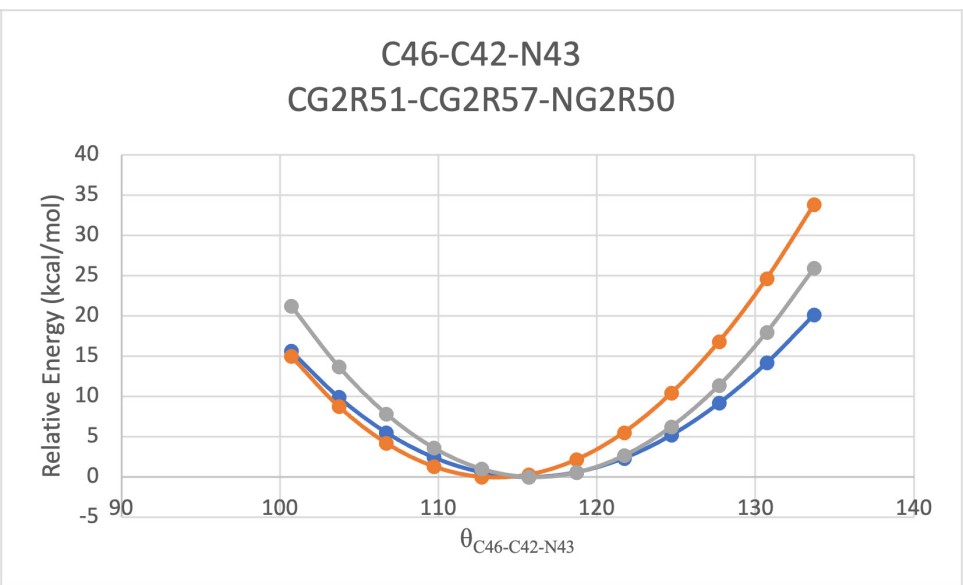

**Appendix 1—figure 2.** Potential-energy curve for the C46-C42-N43 bond angle in RY785 between 100° and 135°, calculated with MP2/6-31G(d) (blue) and with default CGenFF (orange) or with our optimized force field (gray). The Y axis shows the energy (in kcal/mol) relative to the lowest-energy conformer.

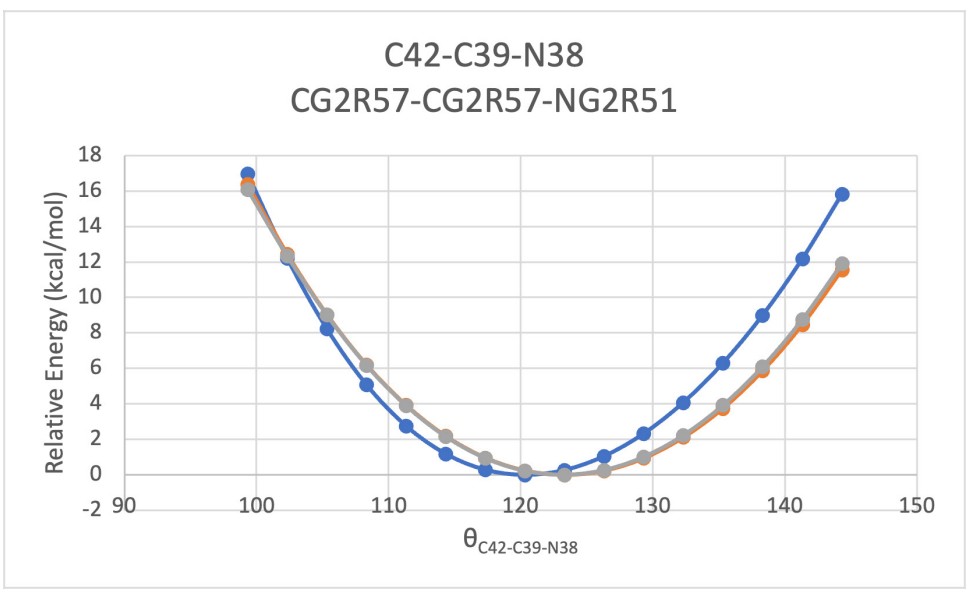

**Appendix 1—figure 3.** Potential-energy curve for the C42-C39-N38 bond angle in RY785 between 100° and 145°, calculated with MP2/6-31G(d) (blue) and with default CGenFF (orange) or with our optimized force field (gray). The Y axis shows the energy (in kcal/mol) relative to the lowest-energy conformer.

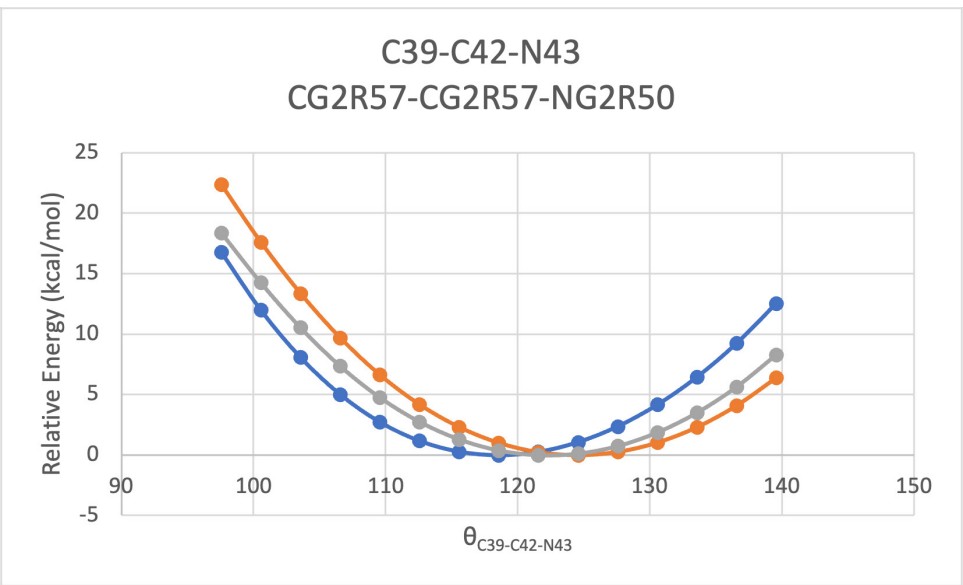

**Appendix 1—figure 4.** Potential-energy curve for the C39-C42-N43 bond angle in RY785 between 95° and 140°, calculated with MP2/6-31G(d) (blue) and with default CGenFF (orange) or with our optimized force field (gray). The Y axis shows the energy (in kcal/mol) relative to the lowest-energy conformer.

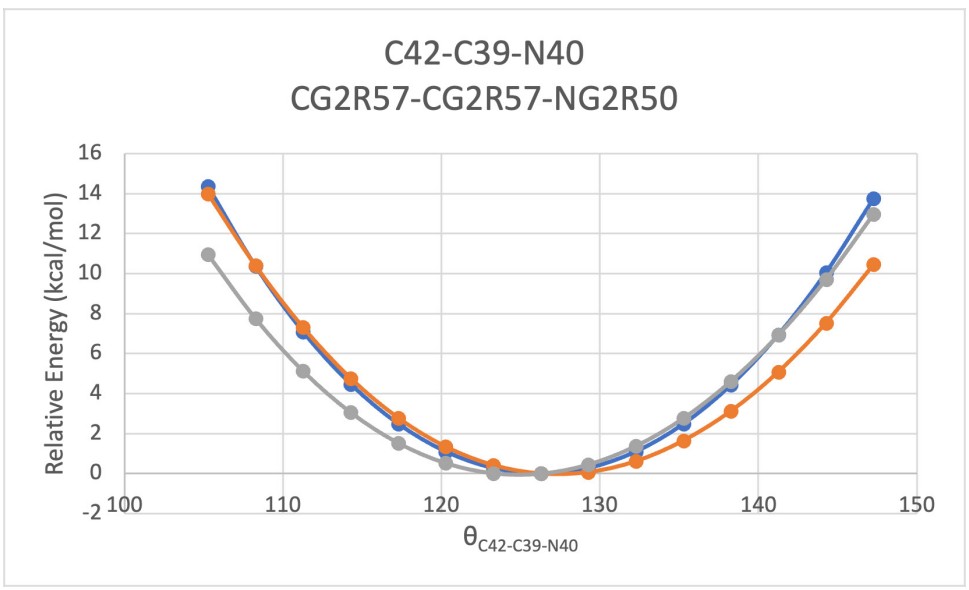

**Appendix 1—figure 5.** Potential-energy curve for C42-C39-N40 bond angle in RY785 between 105° and 150°, calculated with MP2/6-31G(d) (blue) and with default CGenFF (orange) or with our optimized force field (gray). The Y axis shows the energy (in kcal/mol) relative to the lowest-energy conformer.

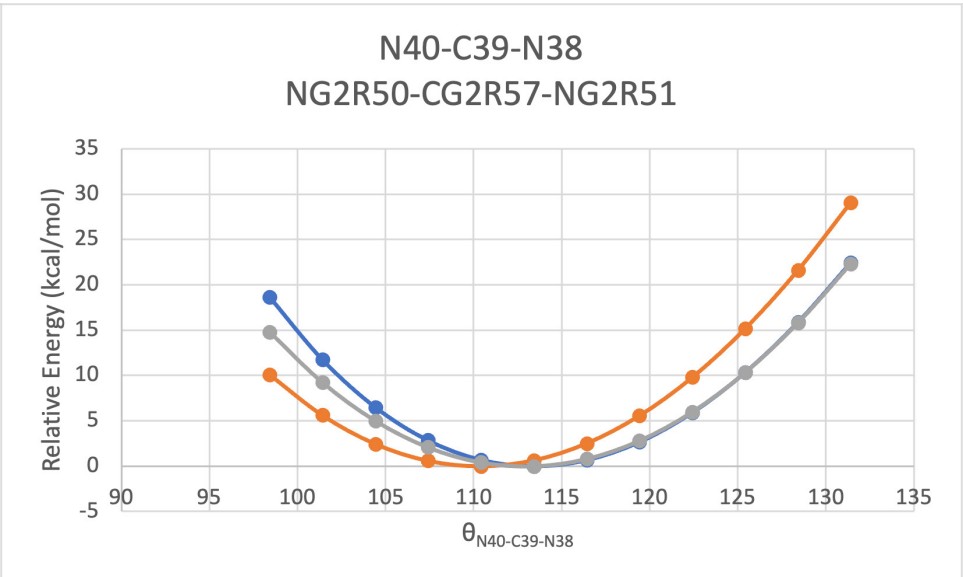

**Appendix 1—figure 6.** Potential-energy curve for the N40-C39-N38 bond angle in RY785 between 95° and 135°, calculated with MP2/6-31G(d) (blue) and with default CGenFF (orange) or with our optimized force field (gray). The Y axis shows the energy (in kcal/mol) relative to the lowest-energy conformer.

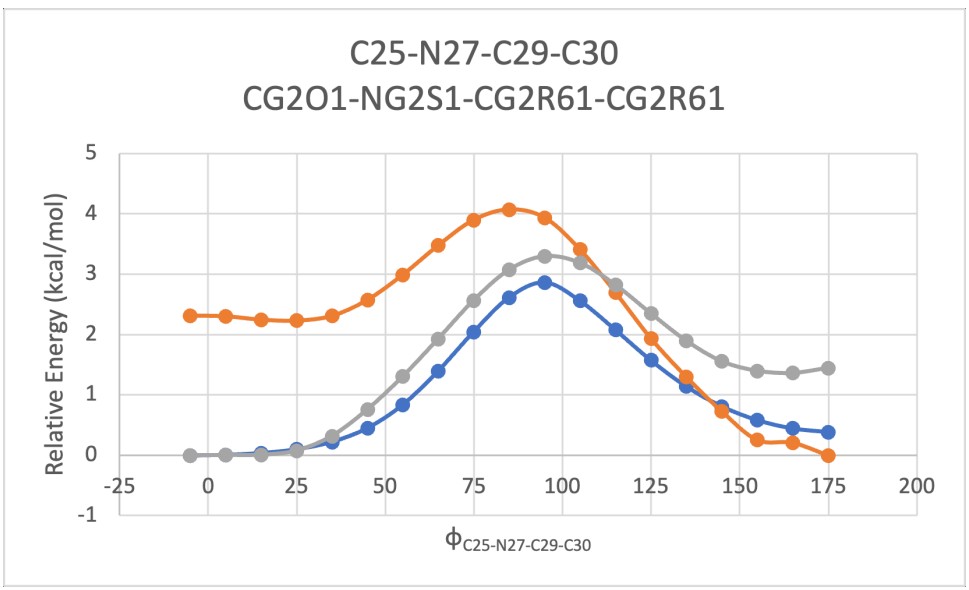

**Appendix 1—figure 7.** Potential-energy curve for the C25-N27-C29-C30 dihedral angle in RY785 between 0° and 180°, calculated with MP2/6-31G(d) (blue) and with default CGenFF (orange) or with our optimized force field (gray). The Y axis shows the energy (in kcal/mol) relative to the lowest-energy conformer.

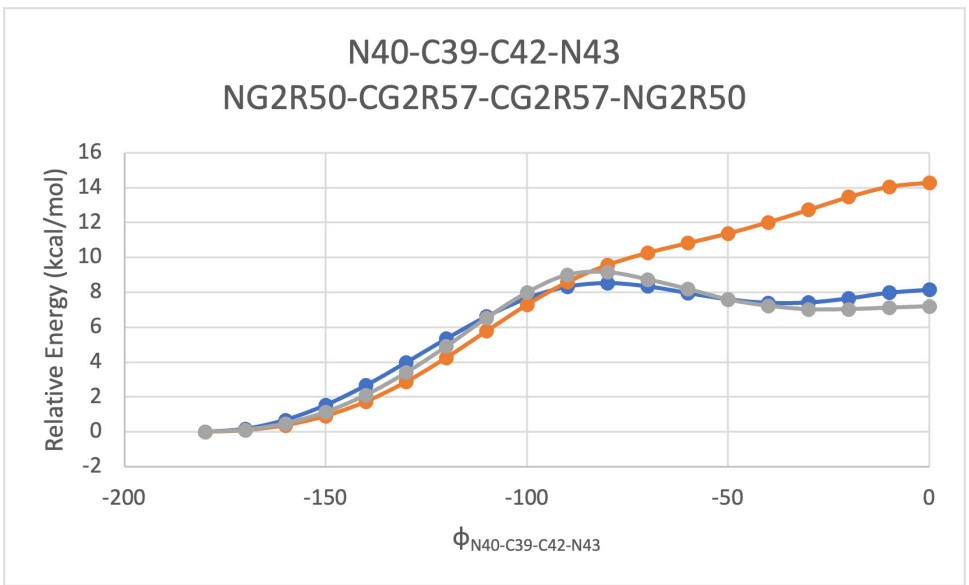

**Appendix 1—figure 8.** Potential-energy curve for the N40-C39-C42-N43 dihedral angle in RY785 between –200° and 0°, calculated with MP2/6-31G(d) (blue) and with default CGenFF (orange) or with our optimized force field (gray). The Y axis shows the energy (in kcal/mol) relative to the lowest-energy conformer.

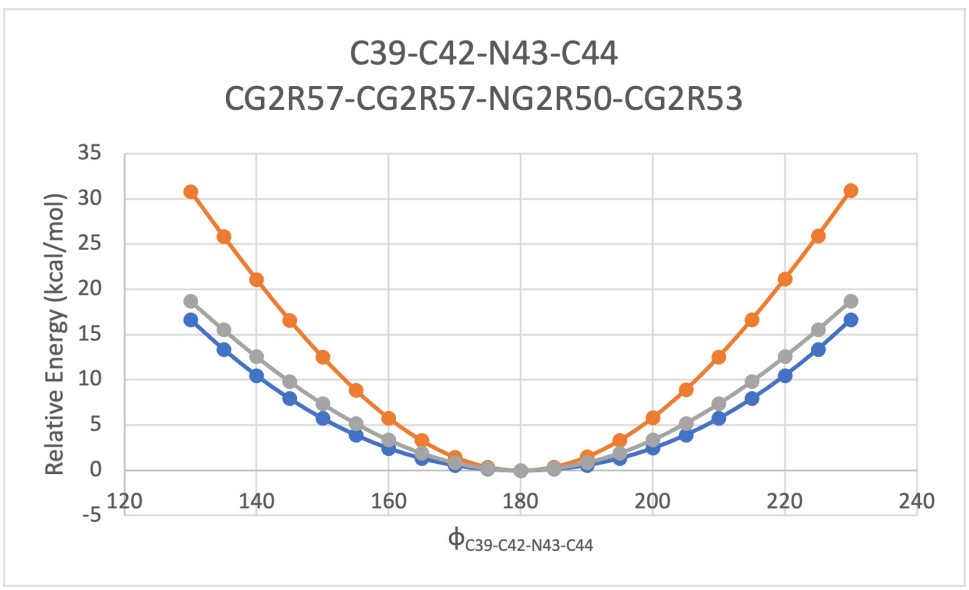

**Appendix 1—figure 9.** Potential-energy curve for the C39-C42-N43-C44 dihedral angle in RY785 between 120° and 240°, calculated with MP2/6-31G(d) (blue) and with default CGenFF (orange) or with our optimized force field (gray). The Y axis shows the energy (in kcal/mol) relative to the lowest-energy conformer.

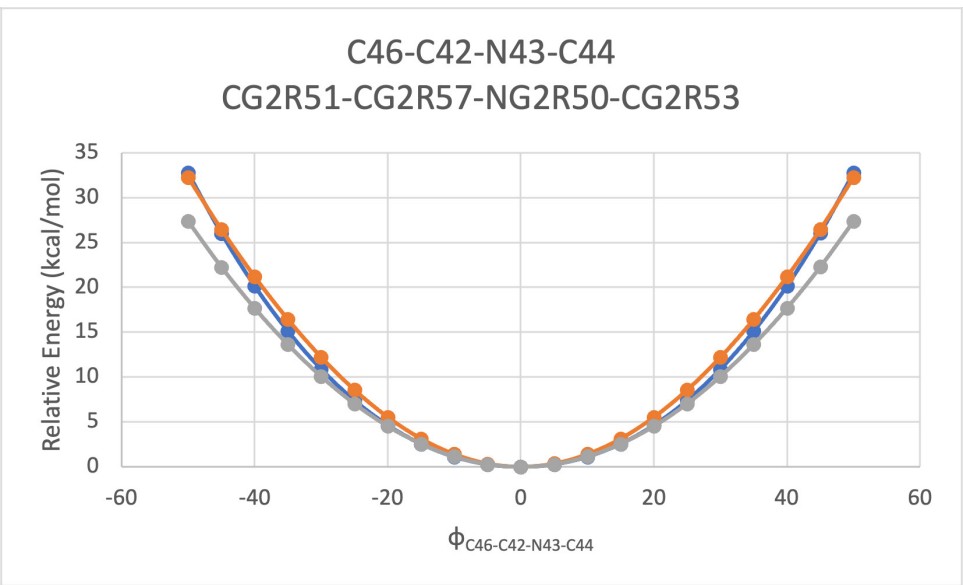

**Appendix 1—figure 10.** Potential-energy curve for the C46-C42-N43-C44 dihedral angle in RY785 between –60° and 60°, calculated with MP2/6-31G(d) (blue) and with default CGenFF (orange) or with our optimized force field (gray). The Y axis shows the energy (in kcal/mol) relative to the lowest-energy conformer.

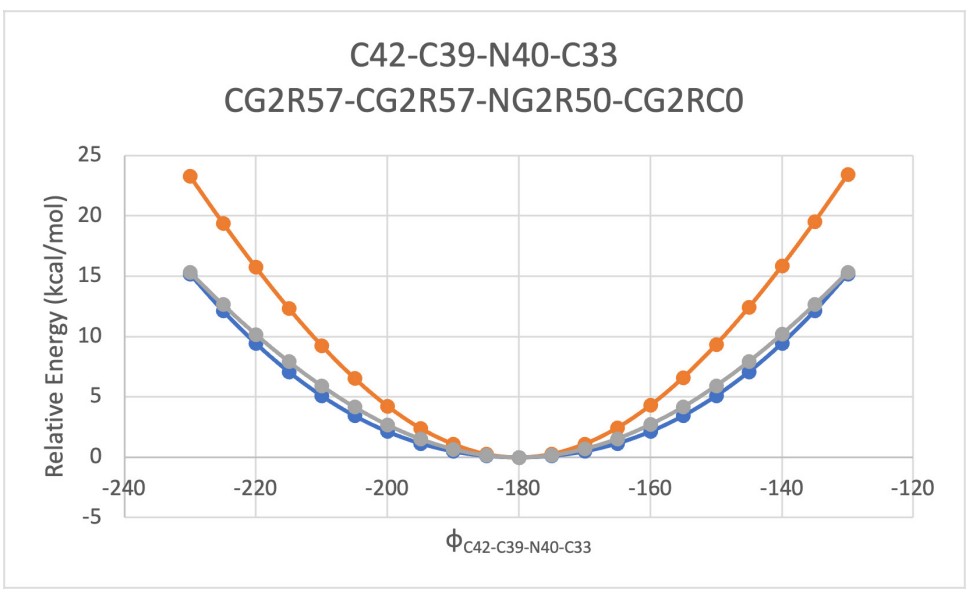

**Appendix 1—figure 11.** Potential-energy curve for the C42-C39-N40-C33 dihedral angle in RY785 between –240° and –120°, calculated with MP2/6-31G(d) (blue) and with default CGenFF (orange) or with our optimized force field (gray). The Y axis shows the energy (in kcal/mol) relative to the lowest-energy conformer.

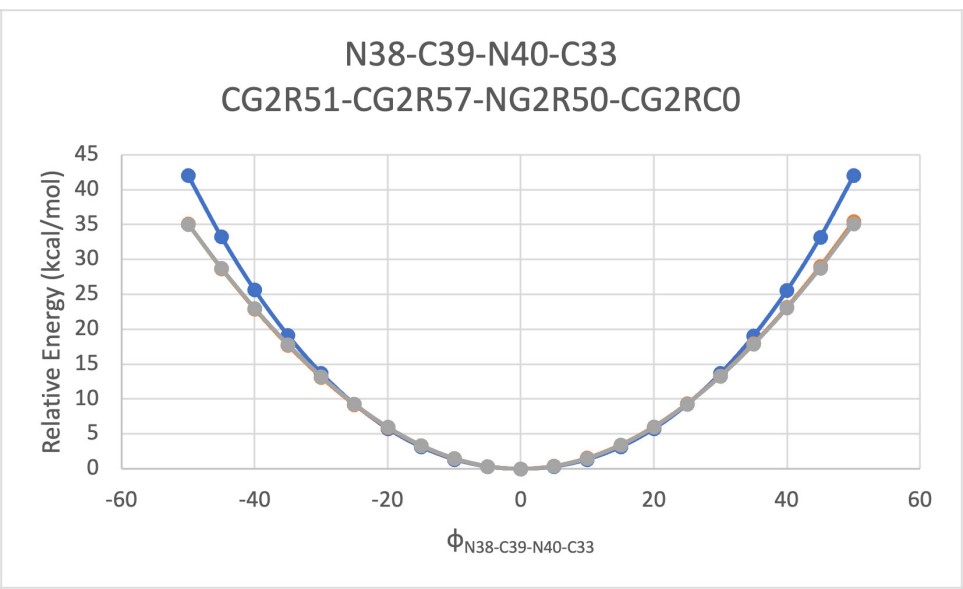

**Appendix 1—figure 12.** Potential-energy curve for the N38-C39-N40-C33 dihedral angle in RY785 between –60° and 60°, calculated with MP2/6-31G(d) (blue) and with default CGenFF (orange) or with our optimized force field (gray). The Y axis shows the energy (in kcal/mol) relative to the lowest-energy conformer.

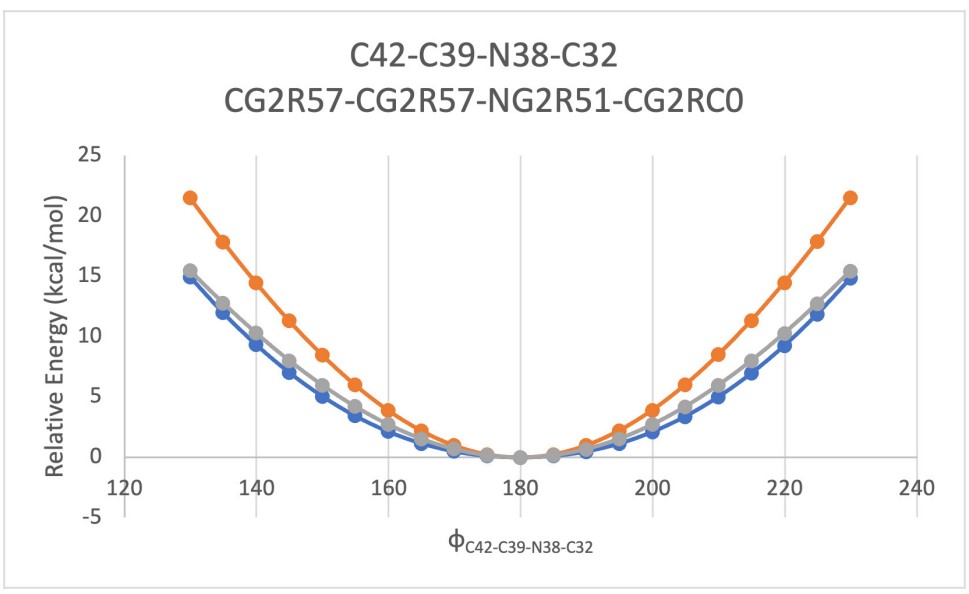

**Appendix 1—figure 13.** Potential-energy curve for the C42-C39-N38-C32 dihedral angle in RY785 between 120° and 240°, calculated with MP2/6-31G(d) (blue) and with default CGenFF (orange) or with our optimized force field (gray). The Y axis shows the energy (in kcal/mol) relative to the lowest-energy conformer.

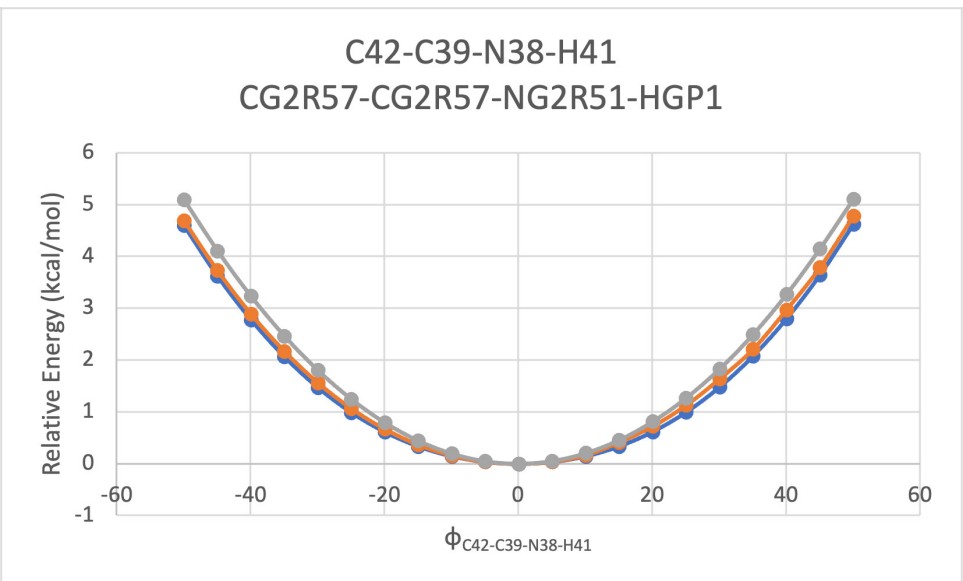

**Appendix 1—figure 14.** Potential-energy curve for the C42-C39-N38-H41 dihedral angle in RY785 between –60° and 60°, calculated with MP2/6-31G(d) (blue) and with default CGenFF (orange) or with our optimized force field (gray). The Y axis shows the energy (in kcal/mol) relative to the lowest-energy conformer.

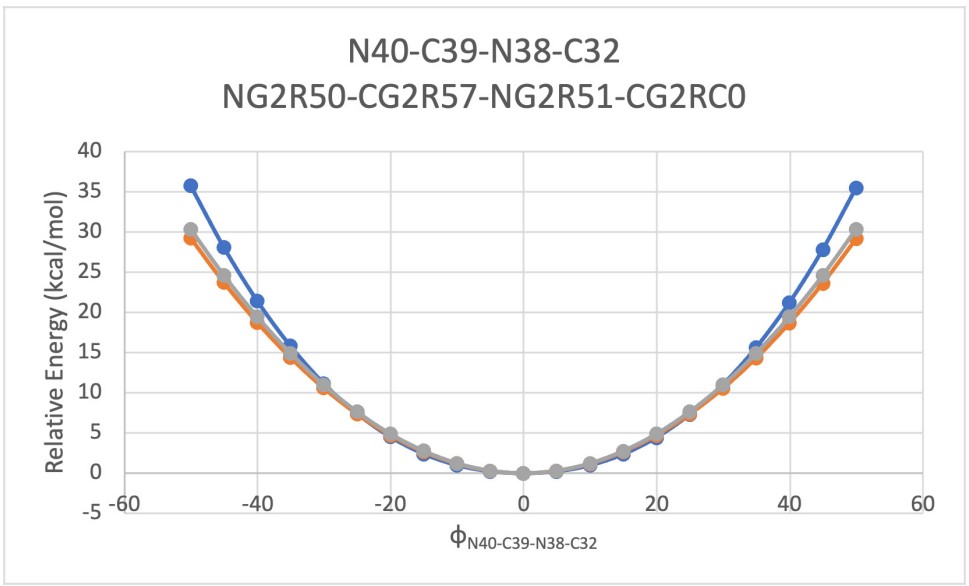

**Appendix 1—figure 15.** Potential-energy curve for the N40-C39-N38-C32 dihedral angle in RY785 between –60° and 60°, calculated with MP2/6-31G(d) (blue) and with default CGenFF (orange) or with our optimized force field (gray). The Y axis shows the energy (in kcal/mol) relative to the lowest-energy conformer.

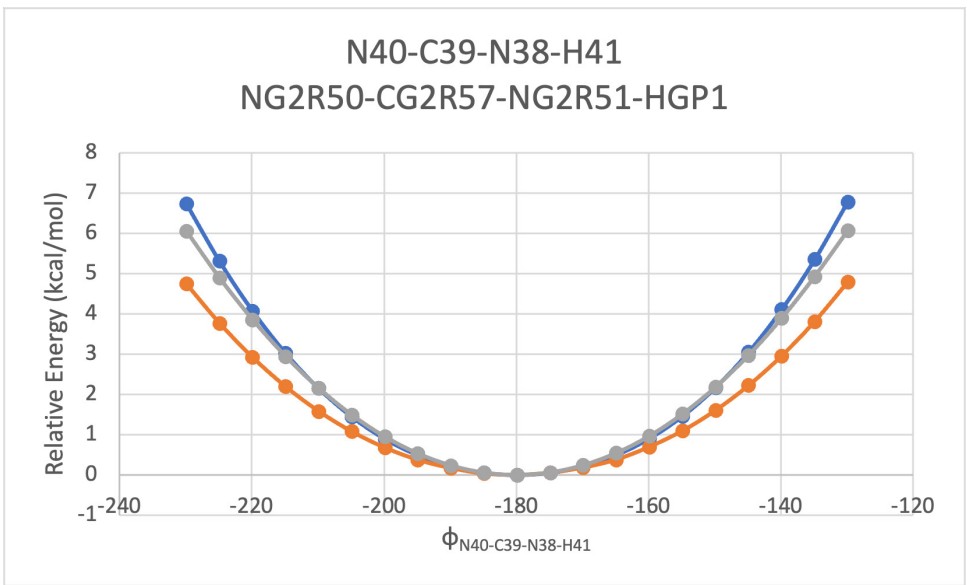

**Appendix 1—figure 16.** Potential-energy curve for the N40-C39-N38-H41 dihedral angle in RY785 between –240° and –120°, calculated with MP2/6-31G(d) (blue) and with default CGenFF (orange) or with our optimized force field (gray). The Y axis shows the energy (in kcal/mol) relative to the lowest-energy conformer.

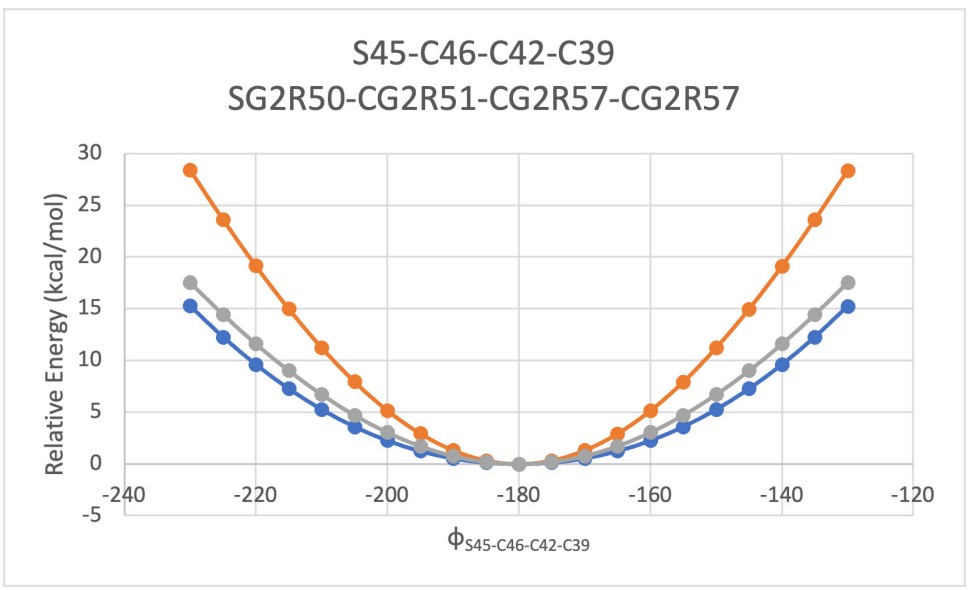

**Appendix 1—figure 17.** Potential-energy curve for the S45-C46-C42-C39 dihedral angle in RY785 between –240° and –120°, calculated with MP2/6-31G(d) (blue) and with default CGenFF (orange) or with our optimized force field (gray) by scanning. The Y axis shows the energy (in kcal/mol) relative to the lowest-energy conformer.

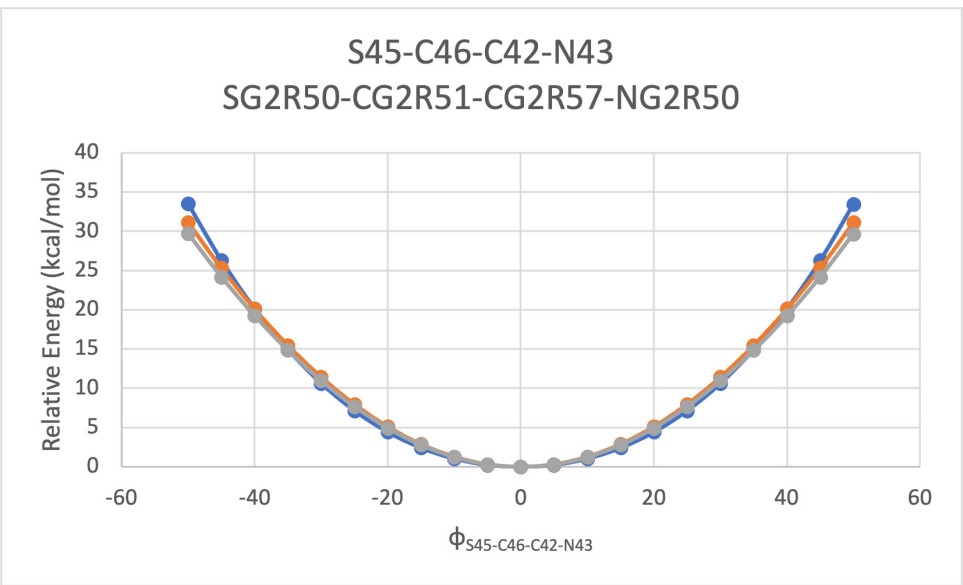

**Appendix 1—figure 18.** Potential-energy curves for the S45-C46-C42-N43 dihedral angle in RY785 between –60° and 60°, calculated with MP2/6-31G(d) (blue) and with default CGenFF (orange) or with our optimized force field (gray). The Y axis shows the energy (in kcal/mol) relative to the lowest-energy conformer.

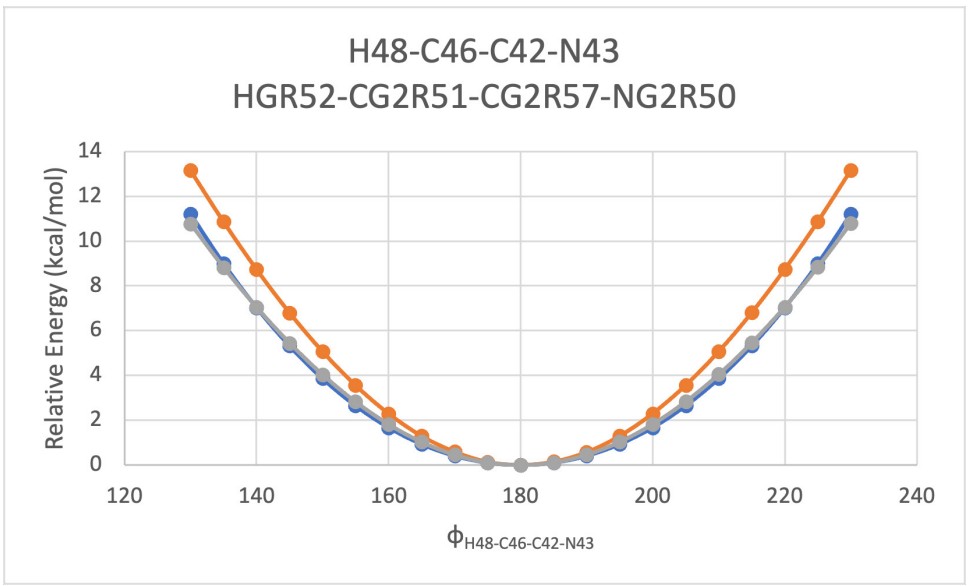

**Appendix 1—figure 19.** Potential-energy curve for the H48-C46-C42-N43 dihedral angle in RY785 between 120° and 240°, calculated with MP2/6-31G(d) (blue) and with default CGenFF (orange) or with our optimized force field (gray). The Y axis shows the energy (in kcal/mol) relative to the lowest-energy conformer.

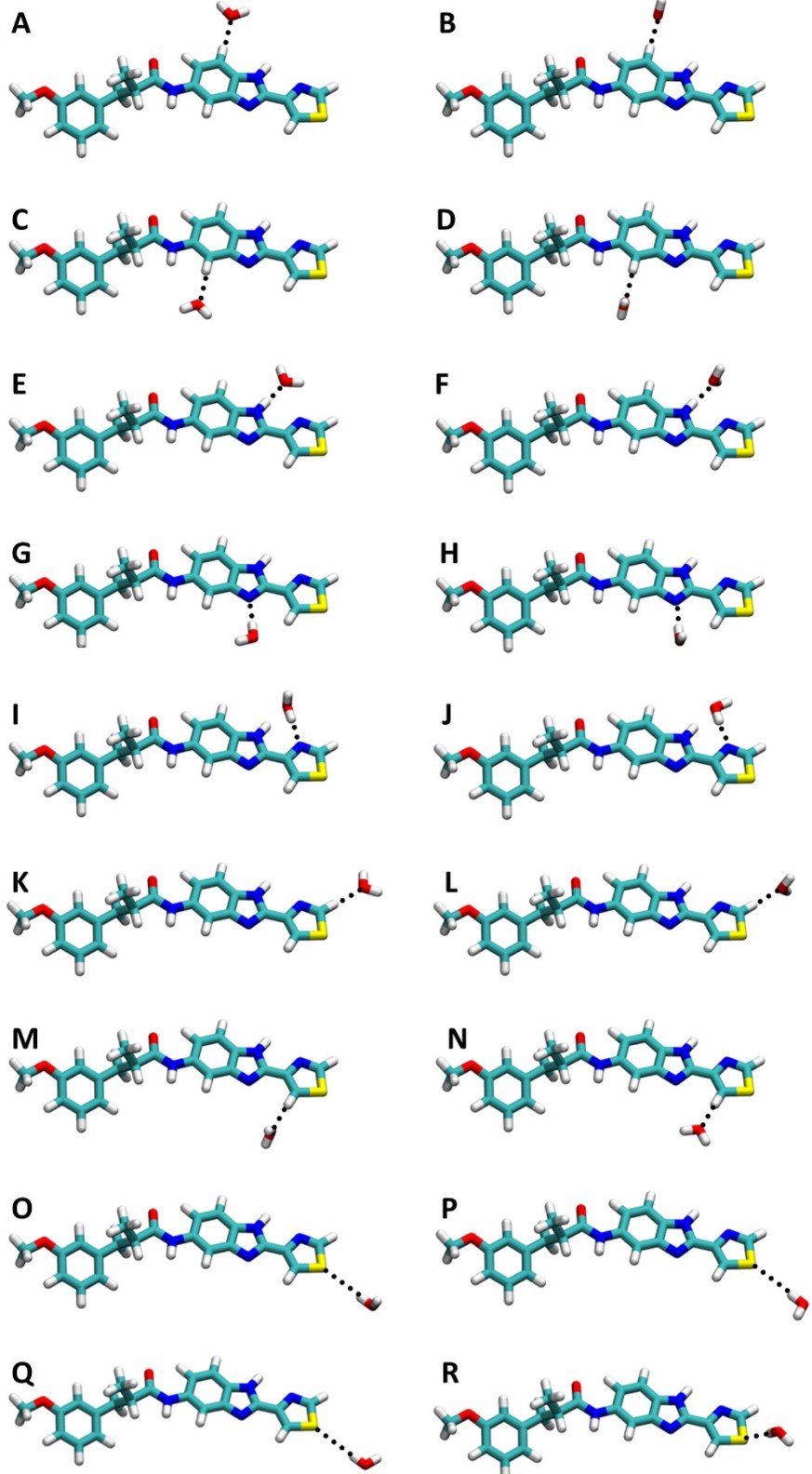

**Appendix 1—figure 20.** Ab initio optimized geometries of RY785-water complexes used in the calibration of our molecular-mechanics force field for RY785.

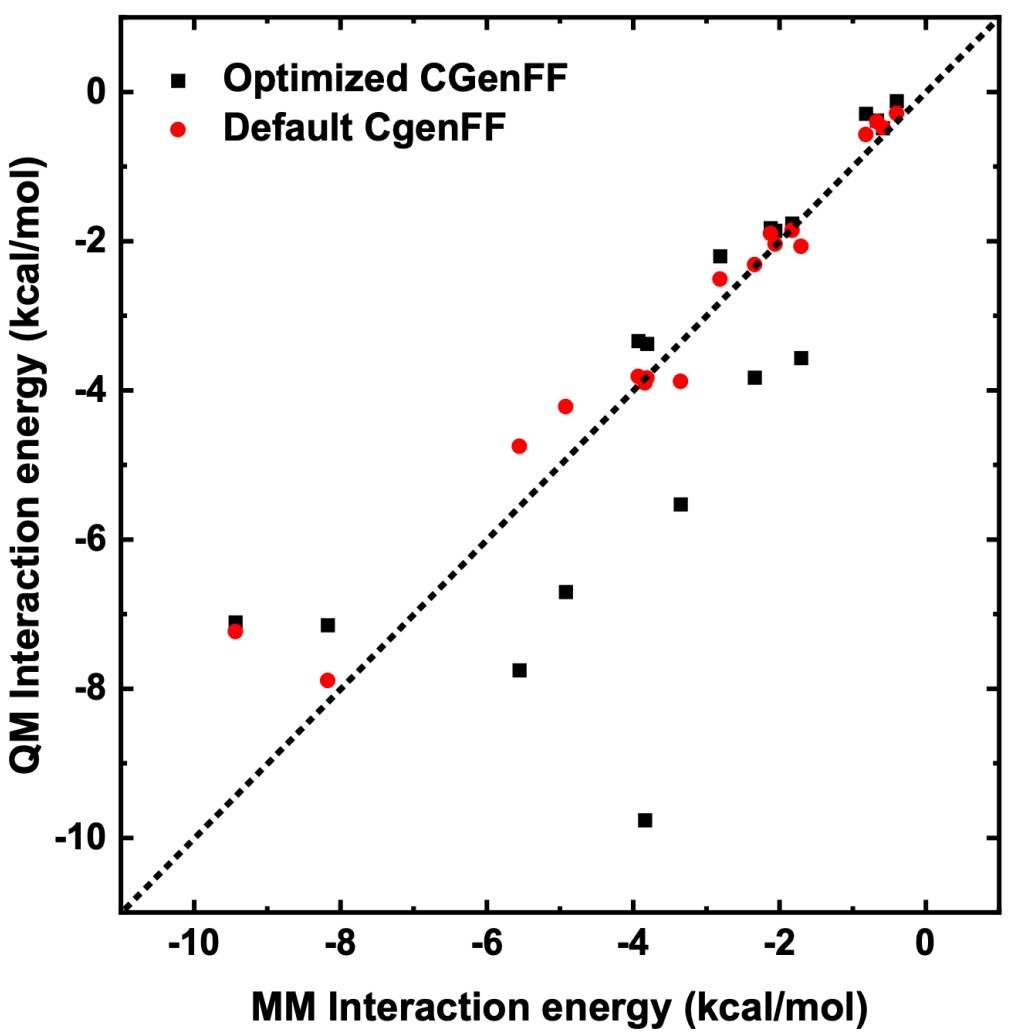

**Appendix 1—figure 21.** Interaction energies (in kcal/mol) for the RY785-water complexes shown in *Appendix 1—figure 20*. QM data (Y-axis) are correlated with MM values calculated using the default CGenFF (red) and our optimized force field (black). The QM interaction energies are calculated with HF/6-31G(d) and scaled by a factor of 1.16. The dashed line represents the equation QM interaction energy = MM interaction energy. The default CGenFF for RY785 results in an average unsigned error of 1.2 kcal/mol as compared to an average unsigned error of 0.3 kcal/mol for the optimized model.

**Appendix 1—table 1.** Interaction distances (in Å) and interaction energies (in kcal/mol) for the RY785-water complexes shown in *Appendix 1—figure 20*.
QM data are compared with MM values calculated using the default CGenFF and our optimized force field.

| Complex | $E_{QM}$ | $r_{QM}$ | $E_{MM, def}$ | $r_{MM, def}$ | $E_{MM, opt}$ | $r_{MM, opt}$ |
|---------|----------|----------|---------------|---------------|---------------|---------------|
| A | −1.83 | 2.58 | −1.76 | 2.60 | −1.86 | 2.59 |
| B | −2.12 | 2.51 | −1.82 | 2.60 | −1.90 | 2.60 |
| C | −1.71 | 2.61 | −3.56 | 2.53 | −2.07 | 2.64 |
| D | −2.34 | 2.47 | −3.83 | 2.51 | −2.32 | 2.62 |
| E | −4.92 | 1.99 | −6.70 | 1.83 | −4.22 | 1.92 |
| F | −3.35 | 2.14 | −5.53 | 1.87 | −3.88 | 1.94 |

*Appendix 1—table 1 Continued on next page*

*Appendix 1—table 1 Continued*

| Complex | $E_{QM}$ | $r_{QM}$ | $E_{MM, def}$ | $r_{MM, def}$ | $E_{MM, opt}$ | $r_{MM, opt}$ |
|---|---|---|---|---|---|---|
| G | −8.17 | 2.06 | −7.15 | 1.88 | −7.89 | 1.87 |
| H | −9.44 | 2.04 | −7.11 | 1.88 | −7.24 | 1.88 |
| I | −5.56 | 2.09 | −7.75 | 1.85 | −3.90 | 1.97 |
| J | −3.84 | 2.07 | −9.76 | 1.83 | −4.75 | 1.95 |
| K | −3.81 | 2.32 | −3.38 | 2.23 | −3.84 | 2.22 |
| L | −3.93 | 2.30 | −3.34 | 2.24 | −3.82 | 2.22 |
| M | −2.06 | 2.31 | −1.86 | 2.26 | −2.04 | 2.25 |
| N | −2.81 | 2.22 | −2.20 | 2.22 | −2.51 | 2.21 |
| O | −0.67 | 2.86 | −0.38 | 2.60 | −0.40 | 2.62 |
| P | −0.59 | 2.92 | −0.49 | 2.59 | −0.48 | 2.60 |
| Q | −0.40 | 2.94 | −0.12 | 2.66 | −0.29 | 2.67 |
| R | −0.82 | 2.80 | −0.29 | 2.58 | −0.37 | 2.57 |

