## [Editor Report · eLife Assessment]

This study represents an **important** advance in our understanding of how certain inhibitors affect the behavior of voltage gated potassium channels. Robust molecular dynamics simulation and analysis methods lead to a new proposed inhibition mechanism with **convincing** strength of support. This study has considerable significance for the fields of ion channel physiology and pharmacology and could aid in development of selective inhibitors for protein targets.

---

## [Referee Report · Reviewer #3 (Public review)]

Summary

In this manuscript, Zhang et al. investigate the conduction and inhibition mechanisms of the Kv2.1 channel, with a particular focus on the distinct effects of TEA and RY785 on Kv2 potassium channels. Using microsecond-scale molecular dynamics simulations, the authors characterize K⁺ ion permeation and RY785-mediated inhibition within the central pore. Their results reveal an inhibition mechanism that differs from those described for other Kv channel inhibitors.

Strengths

The study identifies a distinctive inhibitory mode for RY785, which binds along the channel walls in the open-state structure while still permitting a reduced level of K⁺ conduction. In addition, the authors propose a long-range allosteric coupling between RY785 binding in the central pore and changes in the structural dynamics of Kv2.1. Overall, this is a well-organized and carefully executed study, employing robust simulation and analysis methodologies. The work provides novel mechanistic insights into voltage-gated potassium channel inhibition and may offer useful guidance for future structure-based drug design efforts.

Weaknesses:

As noted in the Discussion, this study focuses primarily on the major binding site within the central pore and was not designed to systematically assess other potential allosteric binding sites for RY785. A more comprehensive structural and biophysical evaluation of possible additional binding sites would be a valuable direction for future investigations.

Comments on revisions:

The authors have addressed my comments.

---

## [Author Response]

The following is the authors’ response to the previous reviews

**Public Reviews:**

**Reviewer #1 (Public review):**
Summary:The authors were seeking to identify a molecular mechanism whereby the small molecule RY785 selectively inhibits Kv2.1 channels. Specifically, the authors sought to explain some of the functional differences that RY785 exhibits in experimental electrophysiology experiments as compared to other Kv inhibitors, namely the charged and non-specific inhibitor tetraethylammonium (TEA). The authors used a recently published cryo-EM Kv2.1 channel structure in the open activated state and performed a series of multi-microsecond-long all-atom molecular dynamics simulations to study Kv2.1 channel conduction under the applied membrane voltage with and without RY785 or TEA present. They observed that while TEA directly blocks K+ permeation by occluding ion permeation pathway, RY785 binds to multiple non-polar residues near the hydrophobic gate of the channel driving it to a semi-closed non-conductive state. They confirmed this mechanism using an additional set of simulations and used it to explain experimental electrophysiology data,Strengths:The total length of simulation time is impressive, totaling many tens of microseconds. The authors develop their own forcefield parameters for the RY785 molecule based on extensive QM based parameterization. The computed permeation rate of K+ ions through the channel observed under applied voltage conditions is in reasonable agreement with experimental estimates of the single channel conductance. The authors have performed extensive simulations with the apo channel as well as both TEA and RY785. The simulations with TEA reasonably demonstrate that TEA directly blocks K+ permeation by binding in the center of the Kv2.1 channel cavity, preventing K+ ions from reaching the SCav site. The authors conclude that RY785 likely stabilizes a partially closed conformation of the Kv2.1 channel and thereby inhibits K+ current. This conclusion is plausible given that RY785 makes stable contacts with multiple hydrophobic residues in the S6 helix, which they can also validate using a recently published closed-state Kv2.1 channel cryo-EM structure. This further provides a possible mechanism for the experimental observations that RY785 speeds up the deactivation kinetics of Kv2 channels from a previous experimental electrophysiology study.Weaknesses:The authors, however, did not directly observe this semi-closed channel conformation and in fact acknowledge that more direct simulation evidence would require extensive enhanced-sampling simulations beyond the scope of this study. They have not estimated the effect of RY785 binding on the protein-based hydrophobic pore constriction, which may further substantiate their proposed mechanism. And while the authors quantified K+ permeation, they have not made any estimates of the ligand binding affinities or rates, which could have been potentially compared to experiment and used to validate their models.However, despite those relatively minor weaknesses, the conclusions of the study are convincing, and overall this is a solid study helping us to understand two distinct molecular mechanisms of the voltage-gated potassium channel Kv2.1 inhibition by TEA and RY785, respectively.
**Reviewer #2 (Public review):**
SummaryIn this manuscript, Zhang et al. investigate the conduction and inhibition mechanisms of the Kv2.1 channel, with a particular focus on the distinct effects of TEA and RY785 on Kv2 potassium channels. Using microsecond-scale molecular dynamics simulations, the authors characterize K⁺ ion permeation and RY785-mediated inhibition within the central pore. Their results reveal an inhibition mechanism that differs from those described for other Kv channel inhibitors.StrengthsThe study identifies a distinctive inhibitory mode for RY785, which binds along the channel walls in the open-state structure while still permitting a reduced level of K⁺ conduction. In addition, the authors propose a long-range allosteric coupling between RY785 binding in the central pore and changes in the structural dynamics of Kv2.1. Overall, this is a well-organized and carefully executed study, employing robust simulation and analysis methodologies. The work provides novel mechanistic insights into voltage-gated potassium channel inhibition and may offer useful guidance for future structure-based drug design efforts.Weaknesses:The study needs to consider the possibility of multiple binding sites for PY785, particularly given its impact on voltage sensors and gating currents. Specifically, the potential for allosteric binding sites in the voltage-sensing domain (VSD) should be assessed, as some allosteric modulators with thiazole moieties are known to bind VSD domains in multiple voltage-gated sodium channels (Ahuja et al., 2015; Li et al., 2022; McCormack et al., 2013; Mulcahy et al., 2019). Increasing structural and functional evidence supports the existence of multiple ligand-binding modes in voltage-gated ion channels. For example, polyunsaturated fatty acids have been shown to bind to KCNQ1 at both the voltage sensor domain and the pore domain (https://doi.org/10.1085/jgp.202012850). Similarly, cannabidiol has been structurally resolved in Nav1.7 at two distinct sites, one in a fenestration and another near the IFM-binding pocket (https://doi.org/10.1038/s41467-023-39307-6). These advances illustrate that ligand effects cannot always be interpreted based solely on a single binding site identified previously.
**Reviewing Editor:**
The comments of the reviewers seem thoughtful and constructive. The weaknesses noted in reviews mainly concern mismatch between expectations, created by reading the Abstract, and data in the manuscript. The mismatch could be reconciled by either new simulations examining a semi-open state of the gate and additional RY785 binding sites, or by adjusting wording of the Abstract and Discussion to make it more clear that such simulations were not done.

The Abstract and Discussion have been revised to make clear the computer-simulations presented in our study were designed to specifically validate or refute the hypothesis that RY785 is recognized by the pore domain, not the voltage sensors.

**Recommendations for the authors:**

**Reviewer #1 (Recommendations for the authors):**
The authors addressed all the major issues in the original submission identified by the reviewers. I noticed a few minor issues, listed below, which can potentially fix small errors and further improve the readability of the manuscript.p.3 tetramethyl-ammonium -> tetraethylammoniump.7 "Snapshot of the final snapshot" -> "Snapshot of the final simulation coordinates"p. 8 "sigma value" - please spell out what it is.p. 9 "one or other subunit of the tetramer" -> "one or another subunit of the tetramer" or "one or more subunits of the tetramer"p 15 "(the net charge of these constructs is thus zero)." -> ""(the net charge of these constructs is zero for these systems)." Please note that using ionizable amino acid residues in their default protonation state does not guarantee net zero charge of the system since the number of cationic and anionic residues is generally not the same.p. 15 "Two K+ ions were initially positioned in the selectivity filter, one coordinated by residues 373..." Please indicate at which ion binding sites S_1, S_2, e.g. K+ were located and what the residue names are .SI Figs. S3-S20. Please indicate in the figure captions that all those data are for RY785SI Fig. S22 and SI Table S1 captions "shown in Fig. S20" -> "shown in Fig. S21"

We thank the Reviewer for this thorough proofreading. We have made the necessary corrections.

**Reviewer #2 (Recommendations for the authors):**
The authors have addressed most of my comments satisfactorily, with the exception of the first point. Below, I provide further clarification regarding my concern.First, it appears that the authors may have misunderstood what is meant by the possibility of multiple binding sites for RY785. This does not imply that the central pore is excluded as a binding site. Rather, it refers to the possibility that, in addition to a pore-domain site, the ligand may interact with additional binding sites, either simultaneously or in a statedependent manner. Increasing structural and functional evidence supports the existence of multiple ligand-binding modes in voltage-gated ion channels. For example, polyunsaturated fatty acids have been shown to bind to KCNQ1 at both the voltage sensor domain and the pore domain (https://doi.org/10.1085/jgp.202012850). Similarly, cannabidiol has been structurally resolved in Nav1.7 at two distinct sites, one in a fenestration and another near the IFM-binding pocket (https://doi.org/10.1038/s41467-02339307-6). These advances illustrate that ligand ecects cannot always be interpreted based solely on a single binding site identified previously. Therefore, even if one assumes that there is no precedent for a small-molecule inhibitor that simultaneously acts on both the voltage sensor and pore domain, this does not exclude the possibility that a ligand may bind to both regions in dicerent functional states.

The Reviewer’s opinion came across clearly in the previous version. We however disagree that a computational investigation of the possibility that RY785 binds to the voltagesensors is well-advised at this point, given that the model we propose seemingly offers a rationale for the inhibitory effects observed experimentally. Our opinion is also that there is no compelling precedent for the mechanism of inhibition envisaged by the Reviewer – and would argue that neither of the two studies referenced above are compelling examples. As we stated in our previous response to the Reviewer, we believe that the logical next step in this research will be to validate or refute the computational prediction we have put forward, experimentally.

In addition, the present computational study does not provide direct mechanistic evidence to explain the statement that RY785 accelerates voltage-sensor deactivation. Specifically, no simulations were performed to model pore-domain closure or voltage-sensor motion upon RY785 binding. Moreover, alternative binding sites were neither explored nor explicitly excluded, as the simulations only involved placing a single molecule of TEA or RY785 approximately 10 Å below the cytoplasmic gate. Under these conditions, conclusions regarding ecects on voltage-sensor dynamics remain speculative.

That is a fair characterization.

These concerns do not detract from the overall quality of this otherwise strong computational study. There are several straightforward ways to address this issue. For example:(1) Perform molecular docking or related screening approaches to evaluate potential ligand-binding sites beyond the central pore, particularly in regions proximal to the voltage sensor. This should not impose a substantial additional computational burden for a computational chemistry group.(2) Revise the abstract and discussion to clarify that the current work focuses exclusively on pore-domain binding and does not explore possible additional binding sites near the voltage sensor. Explicitly stating this limitation would help prevent potential overinterpretation by readers.

We have opted for (2), as noted above.